# DistillNeRF: Perceiving 3D Scenes from Single-Glance Images by Distilling Neural Fields and Foundation Model Features

**Letian Wang**[1,2]    **Seung Wook Kim**[1]    **Jiawei Yang**[3]    **Cunjun Yu**[1,4]    **Boris Ivanovic**[1]

**Steven Waslander**[2]    **Yue Wang**[1,3]    **Sanja Fidler**[1,2]    **Marco Pavone**[1,5]    **Peter Karkus**[1]

[1]NVIDIA Research    [2]University of Toronto    [3]University of Southern California
[4]National University of Singapore    [5]Stanford University

## Abstract

We propose DistillNeRF, a self-supervised learning framework addressing the challenge of understanding 3D environments from limited 2D observations in outdoor autonomous driving scenes. Our method is a generalizable feedforward model that predicts a rich neural scene representation from sparse, single-frame multi-view camera inputs with limited view overlap, and is trained self-supervised with differentiable rendering to reconstruct RGB, depth, or feature images. Our first insight is to exploit per-scene optimized Neural Radiance Fields (NeRFs) by generating dense depth and virtual camera targets from them, which helps our model to learn enhanced 3D geometry from sparse non-overlapping image inputs. Second, to learn a semantically rich 3D representation, we propose distilling features from pretrained 2D foundation models, such as CLIP or DINOv2, thereby enabling various downstream tasks without the need for costly 3D human annotations. To leverage these two insights, we introduce a novel model architecture with a two-stage lift-splat-shoot encoder and a parameterized sparse hierarchical voxel representation. Experimental results on the NuScenes and Waymo NOTR datasets demonstrate that DistillNeRF significantly outperforms existing comparable state-of-the-art self-supervised methods for scene reconstruction, novel view synthesis, and depth estimation; and it allows for competitive zero-shot 3D semantic occupancy prediction, as well as open-world scene understanding through distilled foundation model features. Demos and code will be available at https://distillnerf.github.io/.

## 1 Introduction

Understanding and interpreting complex 3D environments from limited 2D observations is a fundamental challenge in autonomous driving and beyond. Many efforts have been made to tackle this challenge by learning from labor-intensive and costly 3D annotations, such as 3D bounding boxes [1, 2] and semantic occupancy labels [3, 4, 5, 6]. However, these approaches typically struggle with scalability due to their excessive reliance on expensive annotations.

Neural scene representations, such as NeRFs [7, 8] and 3D Gaussian Splatting (3DGS) [9], have recently emerged as a compelling paradigm for learning 3D representations from 2D signals in a self-supervised manner. While these methods demonstrated strong capabilities in view synthesis for indoor scenes [10], and more recently also for challenging dynamic outdoor scenes [11, 12, 13, 14], they require delicately training a new representation for each new scene, leading to extensive computation budget and time needs, typically in the order of hours or minutes. This falls short of the real-time computational requirements for autonomous driving, which typically demand a processing speed of 2-10 Hz. Additionally, most works focus on view synthesis only [11, 12, 13], resulting in learned 3D representations that lack semantics, sidestep downstream tasks, and do not fully exploit the potential

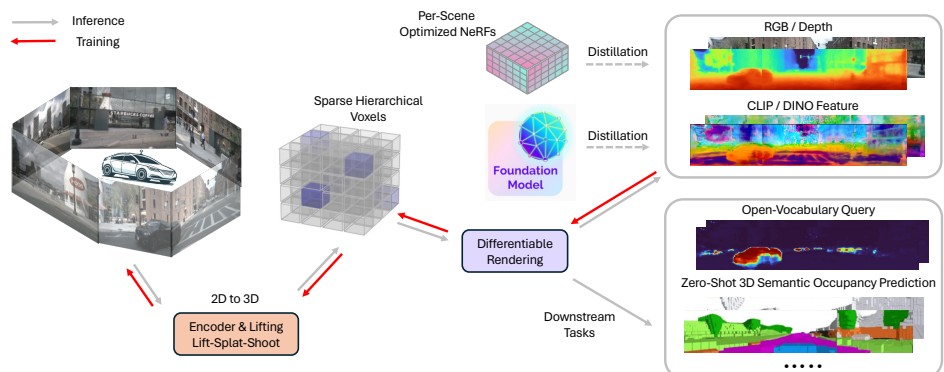

Figure 1: DistillNeRF is a generalizable model for 3D scene representation, self-supervised by natural sensor streams along with distillation from offline NeRFs and vision foundation models. It supports rendering RGB, depth, and foundation feature images, without test-time per-scene optimization, and enables zero-shot 3D semantic occupancy prediction and open-vocabulary text queries.

of neural scene representations to transform various sources of 2D information into 3D, such as features from 2D visual foundation models [15, 16].

To this end, we propose DistillNeRF (Fig. 1), a conceptually simple framework for learning generalizable neural scene representations in driving scenes, by distilling 1) offline optimized NeRFs for enhanced geometry and appearance, and 2) pre-trained 2D foundation models, DINOv2 [17, 16] and CLIP [15], for enriched semantics. Generalizable scene representations with feed-forward models is an active area of research [18, 19, 20], and the autonomous driving domain remains particularly challenging due to sparse camera views with little overlap: prior generalizable NeRFs for driving are shown to be struggling with view synthesis [21, 22, 23]. In contrast, we show that by distilling per-scene optimized NeRFs and visual foundation models, DistillNeRF allows predicting a 3D feature volume with strong geometry, appearance, and semantics, from sparse single-timestep images. The representation is capable of rendering tasks without per-scene optimization (e.g. scene reconstruction, novel view synthesis, foundation model feature prediction), and support downstream tasks, such as open-vocabulary text query and zero-shot 3D semantic occupancy prediction.

DistillNeRF comprises two stages: offline per-scene NeRF training, and distillation into a generalizable model. The first stage trains a NeRF for each scene individually from each scene's driving log, exploiting all available multi-view, multi-timestep information. Specifically, we use EmerNerf [14], a recent NeRF approach with decomposed static and dynamic fields. The second stage trains a generalizable encoder to directly lift multi-camera 2D images captured at a single timestep to a 3D continuous feature field, from which we render images, and supervise with dense depth and novel-view RGB targets generated from the per-scene optimized NeRFs, and foundation model features. Specifically, we propose a novel model architecture with 1) a two-stage Lift-Splat-Shoot encoder [24] to lift 2D observations into 3D; 2) a sparse hierarchical 3D voxel for efficient runtime and memory, parameterized to account for unbounded driving scenes; 3) feature image generation via differentiable volumetric rendering, decoded into appearance, and optionally, foundation model features.

Extensive experiments on the NuScenes [25] and Waymo NOTR [26, 14] dataset demonstrate that DistillNeRF allows for high-quality scene reconstruction and novel view synthesis in previously unseen environments without per-scene training, on par with test-time per-scene optimization approaches, and significantly outperforming previous generalizable approaches. We also show strong results for zero-shot 3D semantic occupancy prediction, and promising quantitative results for open-vocabulary scene understanding.

## 2   Related Work

**Neural Scene Representations.** Neural scene representations, like NeRFs [7, 27] and 3DGS [9, 28], have brought unprecedented success in learning powerful representations of 3D scenes, and have also been successfully applied to challenging driving scenes populated with dynamic objects [29, 30,

12, 31, 32, 33, 34, 35]. However, these methods typically require expensive training for each scene, typically in the order of hours or minutes.

**Generalizable NeRFs.** Generalizable Neural Radiance Fields, such as [18, 36, 37, 38, 39], adapt the capabilities of conventional NeRFs for 3D scene reconstruction and novel view synthesis into a generalizable feedforward model. They replace the costly per-scene optimization with a single feedforward pass through the models. Recent works have extended such approaches to challenging driving scenes and demonstrated the potential in down-stream tasks [22, 5, 23, 40, 41]. However, due to the challenging sparse-view limited-overlapping camera settings on vehicles, these methods usually fail to show strong rendering performance. To the best of our knowledge, we are the very first work to achieve strong scene reconstruction and reasonable novel-view synthesis on par with offline per-scene optimized NeRFs in driving scenes.

**NeRFs with Feature Fields.** Recent advancements extend NeRFs beyond novel view synthesis by integrating 2D features from foundation models into 3D space, equipping neural fields with semantic understanding [42]. Recent approaches also demonstrate similar capabilities in outdoor driving scenes [14] by distilling DINO features into the scene representation. However, these approaches suffer from prolonged optimization times when combined with feature distillation, and thus are impractical for online autonomous driving due to their costly per-scene optimization. Closely related to our work, FeatureNeRF [20] is a generalizable method that distills DINOv2 [17] features into 3D for keypoint transfer, but only investigates simple indoor object-level synthetic datasets like ShapeNet [43], where a large number of overlapping camera images are placed around the object to scan it from various angles. In contrast, we address the more challenging outdoor scene-level settings of autonomous driving, with sparse limited-overlapping camera image inputs. Our method, DistillNeRF, infers 3D feature fields in a single forward pass, making real-time application possible.

**NeRFs in Driving.** Most NeRF-related works in autonomous driving focus on 1) offline scene reconstruction or sensor simulation [11, 12, 14, 13], that accurately reconstruct 3D or 4D scenes with detailed appearance and geometry; 2) exploring potential in downstream tasks [21, 23, 22, 40], that uses volumetric rendering to learn 3D representations from sensor inputs to enlighten online driving. Our work takes the best of these two lines of research: we distill the precise geometry and novel views from offline per-scene optimized NeRFs and rich semantic features from foundation models into our online model. Consequently, our online model not only excels in scene reconstruction and novel view synthesis, but also shows competitive downstream performance, such as zero-shot semantic occupancy prediction, and open-vocabulary query. To the best of our knowledge, we are the first to do so.

**Distilling NeRFs.** Model distillation is a well-established idea [44]. NeRFs have also been distilled into, e.g., Generative Adversarial Networks in [45], and feed-forward models for temporal object shape prediction in [46]. However, prior work mainly focuses on static, object-centric, or indoor scenes. To the best of our knowledge, we are the first to propose distilling a per-scene optimized NeRFs with static-dynamic decomposition into a generalizable model for outdoor driving scenes.

## 3 Method

DistillNeRF predicts a generalizable scene representation in the form of sparse hierarchical voxels from single-timestep multi-view RGB image inputs, and is self-supervised by natural sensor streams, through volumetric rendering to output RGB, depth, and feature images.

The method is depicted in Fig. 1, the detailed architecture in Fig. 2, and key capabilities in Fig. 3 and Fig. 4. Inputs are $N$ posed RGB camera images $\{I_i\}_{i=1}^N$. We use a 2D backbone to extract $N$ feature images $\{X_i\}_{i=1}^N$. We then lift the 2D features to a 3D voxel-based neural field $\mathcal{V} \in \mathbb{R}^{H \times W \times D \times C}$, and apply sparse quantization and convolution to fuse features from multiple views. To account for unbounded scenes we use a parameterized neural field with fixed-scale inner voxels, and varying-scale outer voxels contracting the infinite range. Volumetric rendering is performed to supervise the reconstruction of the scene. For better guidance on scene geometry, we "distill" knowledge from offline optimized NeRFs, using rendered dense depth images from original camera views and virtual camera views. Foundation model features, from CLIP or DINOv2, are set as additional reconstruction objectives and thus are also "distilled" into our model to enrich scene semantics. We introduce design principals in the following section, and refer to Appendix A.7 for implementation details.

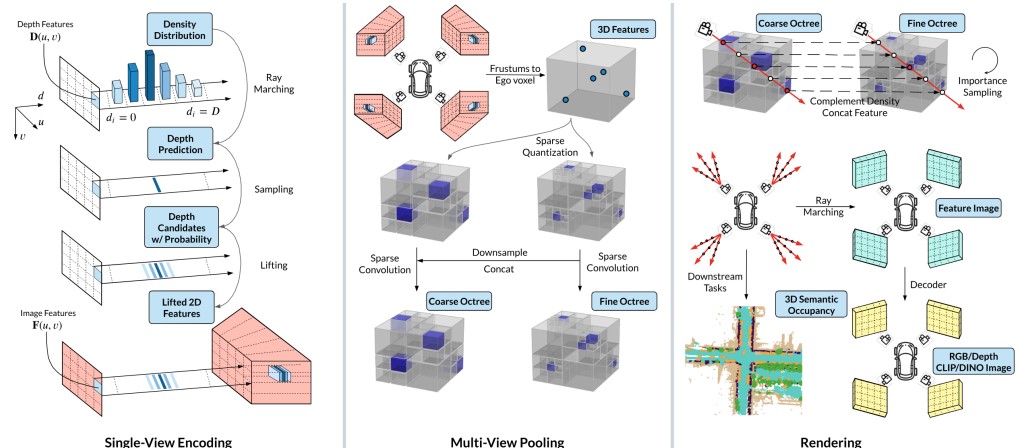

Figure 2: DistillNeRF model architecture. (left) single-view encoding with two-stage probabilistic depth prediction; (center) multi-view pooling into a sparse hierarchical voxel representation using sparse quantization and convolution; (right) volumetric rendering from sparse hierarchical voxels.

## 3.1 Sparse Hierarchical Voxel Model

**Single-View Lifting.** For each of the $N$ camera image inputs, we follow a similar procedure as Lift-Splat-Shoot (LSS) [24] to lift the 2D image features to the 3D neural field. Unlike typical LSS and variants [24, 41, 47] that predict depth in one shot, we propose a two-stage, coarse-to-fine strategy with two jointly trained predictors to capture more nuanced depth. The first stage, following prior LSS works, predicts categorical depth and aggregates them into a single prediction with ray marching. The second stage then predicts a distribution over a fine-grained set of categorical depth values, which are centered around the first-stage predicted coarse depth.

Specifically, in the first stage, we feed each image to a 2D backbone to generate a depth feature map of size $H \times W \times D$. The depth feature map is regarded as a discrete frustum where $D$ denotes the number of pre-defined categorical depths. Inspired by the volume rendering equation [7], each entry in the frustum is a density value. That is, the $d$'th channel of the frustum at pixel $(h, w)$ represents the density value $\sigma_{h,w,d}$ of the frustum entry at $(h, w, d)$. The occupancy weight of entry $(h, w, d)$ is then

$$\mathbb{O}(h, w, d) = exp(-\sum_{j=1}^{d-1} \delta_j \sigma_{h,w,j})(1 - exp(-\delta_d \sigma_{h,w,d})), \quad (1)$$

where $\delta_d = t_{d+1} - t_d$ is the distance between each pre-defined depth $t$ in the frustum. Coarse depth for pixel $(h, w)$ is obtained by aggregating with ray marching:

$$\mathbb{D}(h, w) = \sum_{d=1}^{D} \mathbb{O}(h, w, d) t_d. \quad (2)$$

In the second stage, centered around the initial coarse depth prediction, we dynamically sample a set of $D'$ fine-grained depth candidates. This involves uniform sampling, with the sampling range adaptively adjusted based on the coarse depth estimate. We then embed these fine-grained depth candidates, and combine their embeddings with the depth features from the first stage, and feed them to another network to generate the density of each fine-grained depth candidate. The occupancy weights $\mathbb{O}'$ of the fine-grained depth candidates are predicted similarly by Eq 1, which can also be regarded as probabilities of each fine-grained depth candidate.

With the candidate depths associated with probabilities, we then lift 2D image features to 3D. Specifically, we use a 2D image backbone to get 2D image features $\phi$, and assign the 2D image features to the 3D frustum according to each pixel's depth. That is, for pixel $(h, w)$, its image feature $\phi_{h,w}$ is distributed to each fine-grained depth candidates $t'_d$ by $[\mathbb{O}'_{h,w,d}\phi_{h,w}, \sigma'_{h,w,d}]$, where we scale the pixel image feature $\phi_{h,w}$ with occupancy $\mathbb{O}'_{h,w,d}$ and concatenate it with density $\sigma'_{h,w,d}$.

**Multi-View Fusion.** After constructing the frustum for each view, we transform the frustums to the world coordinates using the camera poses, and fuse them into a shared 3D voxel-based neural field $\mathcal{V}$,

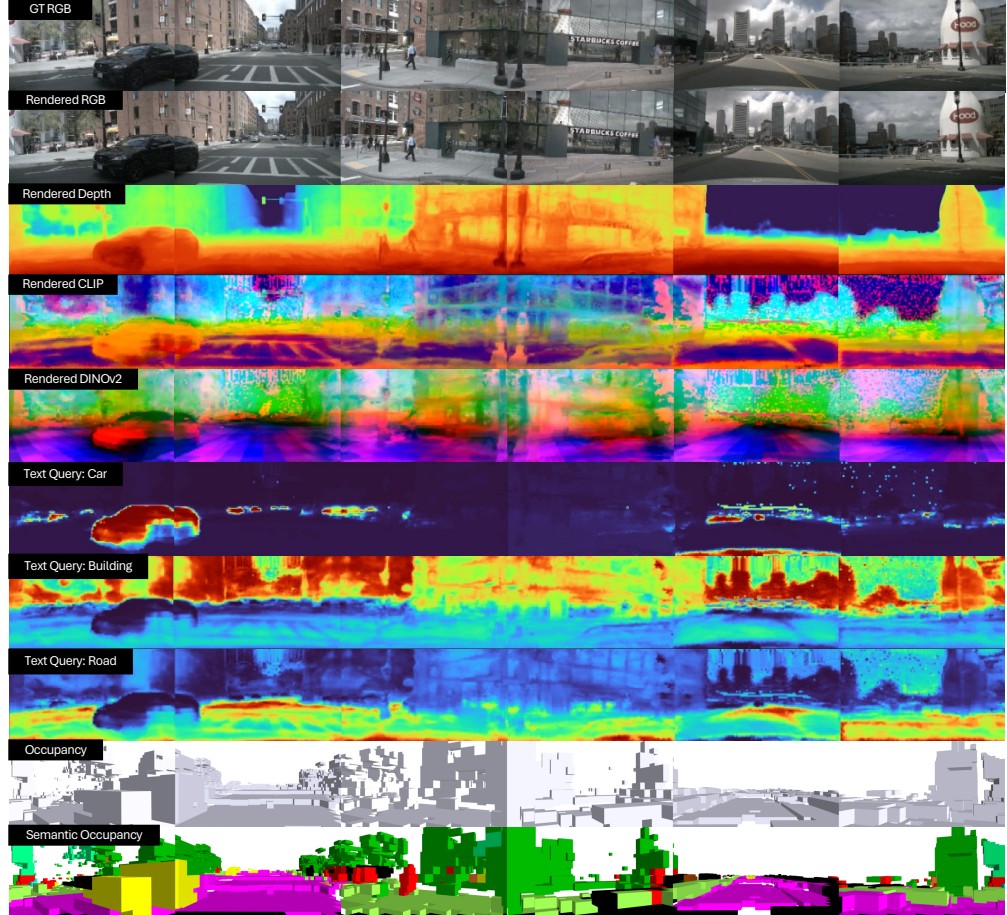

Figure 3: DistillNeRF Capabilities - Given single-frame multi-view cameras as input and without test-time per-scene optimization, DistillNeRF can reconstruct RGB images (row 2), estimate depth (row 3), render foundation model features (rows 4, 5) which enables open-vocabulary text queries (rows 6, 7, 8), and predict binary and semantic occupancy in zero shot (rows 9, 10).

where each voxel represents a region in the world coordinates and carries both densities and features. When lifted frustum entries from different views lie in the same voxel, we fuse them with average pooling.

**Sparse Hierarchical Voxels.** Unlike previous works [41, 48, 21] using dense voxels, which uniformly quantizes the neural field and potentially wastes computation and memory on large empty regions, we apply sparse quantization on the neural field. Specifically, we follow the octree representation [49] to recursively divide the neural field into specified levels, according to the 3D positions of the lifted 2D features. While an octree with many levels and thus smaller voxel sizes can capture more accurate 3D positions of lifted features, overly fine-grained octrees can lead to difficulty in querying features during rendering (e.g. missing far-away features due to large gaps between sampled rays). To this end, we generate two octrees with different quantization granularities, one fine octree with more quantization levels capturing details of the lifted features, and one coarse octree to represent general information of a larger range. Sparse convolutions [50] are then applied to both octrees to encode the relationships and interactions among voxels, during which the features in the fine octree are also downsampled and concatenated with the coarse octree to enhance details.

**Neural Field Parameterization.** Unlike prior works that consider a neural field covering a fixed range [41, 21, 22], our work aims at accounting for the unbounded-scene settings in the driving scenes by proposing a parameterized neural field. We want to keep the inner-range voxels at the real scale and high resolution due to their importance to various downstream tasks (e.g., occupancy prediction in 50 meters' range), while contracting the scene up to infinite distance in the outer range of the voxels at a lower resolution, so we can render with low memory and computation cost (e.g. sky, far-away buildings). Inspired by [8, 51], we propose a transform function that maps a 3D point

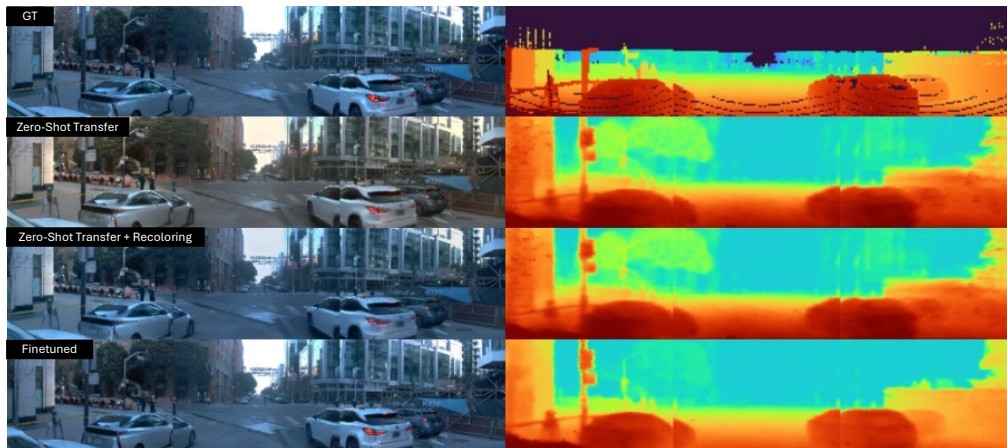

Figure 4: DistillNeRF Generalizability - Trained on the nuScenes dataset, our model demonstrates strong zero-shot transfer performance on the unseen Waymo NOTR dataset, achieving decent reconstruction quality (row 2). This quality can be further enhanced by applying simple color alterations to account for camera-specific coloring discrepancies (row 3). After fine-tuning (row 4), our model surpasses the offline per-scene optimized EmerNeRF, achieving higher PSNR (29.84 vs. 28.87) and SSIM (0.911 vs. 0.814). See Tab 3 for quantitative results.

in the world coordinates $p = (x, y, z)$ to the coordinates in the parameterized neural field:

$$f(p) = \begin{cases} \alpha \frac{p}{p_{inner}} & \text{if } |p| \le p_{inner} \\ \left(1 - \frac{p_{inner}}{|p|}(1 - \alpha)\right) \frac{p}{|p|} & if\, |p| > p_{inner} \end{cases}. \tag{3}$$

The transformed coordinates $f(p)$ will always be within $[0, 1]$, where $p_{inner}$ sets the range of the inner voxel (region of interest) and varies in $x, y, z$ directions, and $\alpha \in [0, 1]$ denotes the contraction ratio, namely the proportion of the inner range in the parameterized neural field. Consistent parameterizations are enforced for both the single-view lifting process (on the depth space) and the multi-view fusion process (on the 3D coordinate space).

**Volume Rendering from Sparse Hierarchical Voxels.** Finally, we use differentiable volumetric rendering to project the 3D neural field onto 2D feature maps and render images. Specifically, for each pixel of each camera, we shoot a ray originating from the camera to the neural field according to the camera poses, and sample points along the ray. *Feature Querying:* For each sample point, we query both fine and coarse octree to get the density and features of the corresponding voxel that the sample point lies in. Further, to capture both high-level information and fine-grained details, the features from both octrees are concatenated as the final feature. *Density Querying:* Regarding the density, while the fine octree captures more accurate 3D positioning, since the fine octree voxel only covers a small region, the sample points could be easily within empty voxels and thus query no information, especially for faraway regions. To this end, for each sample point, we first query the fine octree to get the fine density. If the fine density is empty, we query the coarse octree to complement the density. *Two-Stage Sampling:* Regarding the sampling strategy, we follow [8] to sample points for each ray with two phases: first we sample a set of points uniformly, then we sample another set of points with importance sampling given densities for the first set of points, so to enhance surface details in the scene. With the densities and features of the sampled points we do volumetric rendering using Eq 2 to get the 2D feature map for each camera. *Decoding:* The rendered 2D feature maps are then fed into a CNN decoder to enhance high-frequency details, and upsample the final RGB image without increasing the rendering resolution/cost. Note that from the volume rendering process, we can also get the expected depth for each pixel [29].

### 3.2 Self-supervised Training with Distillation

**Distillation from Offline NeRFs.** While our model can be trained by simply reconstructing RGB images, it remains challenging to learn scene geometry from only single time-step camera image inputs. The challenge is especially pronounced with typical autonomous vehicle setups where mounted cameras are facing outwards and have limited view overlap, making multi-view recon-

struction degrade to the monocular setting and aggravating depth ambiguity. A natural idea is to use images from multiple time steps to encourage view overlaps. However, driving scenes typically contain many dynamic objects that move between time steps, introducing noise to the reconstruction objective. Instead, we propose to leverage the high-quality geometry of per-scene optimized NeRFs that aggregate information from a full sensor stream. Specifically, we use EmerNeRF [14], a recent NeRF approach that handles dynamic objects by decomposing the scene into static and dynamic fields in a self-supervised manner. We propose two different ways to distill knowledge from per-scene optimized NeRFs, which together construct $L_{NeRF}$, a distillation loss from offline NeRFs:

- **Dense 2D depth.** Depth supervision from LiDAR point clouds, $L_{depth}$, is commonly used to facilitate 3D geometry learning, however, point clouds are typically sparse and only provide depth labels for a limited horizontal/vertical range. Thus we propose to use offline optimized NeRFs as a depth auto-labeling tool. Specifically, for each training target image we render a dense depth map from the offline NeRF, and use it as additional depth supervision, $L_{depth'}$.

- **Virtual cameras.** In addition to depth distillation from original camera views, we can leverage temporally decomposed NeRFs to render depth from "virtual cameras", i.e., novel views, while keeping the time dimension frozen. In this manner, the virtual depth and RGB images can be used as additional reconstruction targets, thus the number of target images and the view overlap between cameras can be artificially increased, encouraging consistent depth prediction and improving novel-view synthesis performance.

**Distillation from Foundation Models.** In addition to RGB and depth prediction, can we learn 3D representations that contain rich semantics and enable a wider range of downstream tasks? Witnessing the rise of vision foundation models with generalized capabilities across various vision tasks, we propose to distill 2D foundation model features, such as CLIP [15] and DINOv2 [16], into our 3D scene representation model. We achieve this by simply introducing an additional MLP to our rendered 2D feature images, and train the model to reconstruct the foundation model feature images with an L1 loss $L_{found}$. We demonstrate early attempts at utilizing these foundation models on open-vocabulary text query tasks shown in Fig 3, but we leave more comprehensive explorations to future work.

**Training objective.** In summary, we train our model for a linear combination of loss terms:

$$L = \underbrace{L_{rgb} + L_{depth} + L_{density}}_{\text{rendering}} + \underbrace{L_{NeRF} + L_{found}}_{\text{distillation}}, \tag{4}$$

where $L_{rgb}$ and $L_{depth}$ are rendering losses for RGB and depth; $L_{density}$ is a density entropy loss encouraging clearer surfaces and structured density values as in [14]; $L_{NeRF}$ and $L_{found}$ denote distillation losses from offline NeRFs and foundation models. Please refer to Appendix A.5 for more details on the losses.

## 4   Experiments

We benchmark DistillNeRF against SOTA generalizable NeRFs, offline NeRFs, as well as comparable methods on the popular NuScenes dataset [52]. We first evaluate the rendering performance, i.e., scene reconstruction, novel view synthesis, and feature reconstruction (Table 1). We further evaluate the learned 3D geometry through depth estimation (Table 2), and 3D semantic occupancy prediction (Table 4). Ablations of DistillNeRF are also analyzed in each task and displayed in each table, additional ablations are available in Tab A.2, Tab 6, and Tab A.6 of the Appendix. Qualitative results, including open vocabulary queries, are in Fig. 3, 6, and 7. Videos are online at https://distillnerf.github.io/. Implementation and training details are in the Appendix A.7.

**Dataset**. The nuScenes dataset [52] contains 1000 driving scenes from different geographic areas, each scene capturing approx. 20 seconds of driving, resulting in approx 40000 frames in total. Scenes are captured via six cameras mounted on the vehicle heading in different directions along with point clouds from LiDAR. We use the default data split, 700 scenes for training, 150 scenes for validation. We adopt the resolution of input RGB images, rendered RGB, and rendered depth are 114×228, 114×228, and 64×114 respectively. We also evaluate the generalizability of our method on the Waymo NOTR dataset [14], a balanced and diverse benchmark derived from the Waymo Open Dataset [26]. NOTR features 120 unique, hand-picked driving sequences, split into 32 static, 32 dynamic, and 56 diverse scenes across seven challenging conditions. We adopt the resolution of input RGB images, rendered RGB, and rendered depth are 144×216, 144×216, and 72×108 respectively.

Table 1: Reconstruction and novel-view synthesis on nuScenes validation set. DistillNeRF is on par with the per-scene optimized NeRFs, both in RGB and foundation feature rendering, and significantly outperforms SOTA generalizable NeRF methods. In the DistillNeRF variants, we denote 'Depth' as the depth distillation from offline NeRFs, 'Param.' as the parameterized space, and 'Virt.' as the distillation from virtual cameras in offline NeRFs. See Fig. 6 and Fig. 7 for qualitative results.

| Method | Single Timestep Input | No Test-Time Per-Scene Opt | Foundation Model Lifting | RGB Reconstruct | | RGB Novel-View Synthesis | | Foundation Feature Reconstruction | |
|---|---|---|---|---|---|---|---|---|---|
| | | | | PSNR ↑ | SSIM ↑ | PSNR ↑ | SSIM ↑ | CLIP PSNR ↑ | DINOv2 PSNR ↑ |
| EmerNerf [14] | ✗ | ✗ | ✔ | **30.88** | 0.879 | - | - | **20.91** | **21.12** |
| Single-Frame EmerNerf | ✔ | ✗ | ✔ | - | - | **20.95** | 0.585 | - | - |
| SelfOcc [22] | ✔ | ✔ | ✗ | 20.67 | 0.556 | 18.22 | 0.464 | - | - |
| UniPad [21] | ✔ | ✔ | ✗ | 19.44 | 0.497 | 16.45 | 0.375 | - | - |

| Depth | Param. | Virt. | | | | DistillNeRF Variants | | | | | |
|---|---|---|---|---|---|---|---|---|---|---|---|
| ✗ | ✗ | ✗ | ✔ | ✔ | ✔ | 28.01 | 0.872 | 19.12 | 0.501 | - | - |
| ✔ | ✗ | ✗ | ✔ | ✔ | ✔ | 30.11 | **0.917** | 20.27 | 0.567 | 18.69 | 18.48 |
| ✔ | ✔ | ✗ | ✔ | ✔ | ✔ | 28.42 | 0.879 | 20.06 | 0.565 | - | - |
| ✔ | ✔ | ✔ | ✔ | ✔ | ✔ | 28.72 | 0.880 | 20.78 | **0.590** | - | - |

Table 2: Depth estimation results on the nuScenes validation set. Depth targets are defined by (a) sparse LiDAR scans or (b) dense depth images rendered from EmerNerf. We use highlighting across comparable methods with rendering support and no test-time optimization. DistillNeRF outperforms comparable generalizable NeRF methods, especially on dense depth targets.

| (a) Sparse LiDAR GT | No Test-Time Per-Scene Opt | Support Rendering | Abs Rel ↓ | Sq Rel ↓ | RMSE ↓ | RMSE log ↓ | $\delta < 1.25$ ↑ | $\delta < 1.25^2$ ↑ | $\delta < 1.25^3$ ↑ |
|---|---|---|---|---|---|---|---|---|---|
| EmerNerf [14] | ✗ | ✔ | 0.073 | 0.346 | 2.696 | 0.159 | 0.942 | 0.975 | 0.986 |
| SelfOcc* [22] | ✔ | ✗ | 0.214 | 2.418 | 6.556 | 0.31 | 0.745 | 0.875 | 0.932 |
| SelfOcc [22] | ✔ | ✔ | 0.342 | 5.497 | 7.678 | 0.370 | 0.705 | 0.841 | 0.905 |
| UniPAD [21] | ✔ | ✔ | 0.254 | 2.945 | 5.903 | 0.318 | 0.670 | 0.867 | 0.935 |

| Depth Distill | Virtual Distill | | | DistillNeRF Variants | | | | | |
|---|---|---|---|---|---|---|---|---|---|
| ✗ | ✗ | ✔ | ✔ | 0.248 | 3.090 | 6.096 | 0.312 | 0.704 | 0.885 | 0.947 |
| ✔ | ✗ | ✔ | ✔ | 0.233 | 2.890 | 5.890 | 0.296 | 0.703 | 0.881 | 0.945 |
| ✔ | ✔ | ✔ | ✔ | **0.223** | **1.776** | **5.461** | **0.293** | **0.763** | **0.903** | **0.961** |

| (b) Dense Depth GT | No Test-Time Per-Scene Opt | Support Rendering | Abs Rel ↓ | Sq Rel ↓ | RMSE ↓ | RMSE log ↓ | $\delta < 1.25$ ↑ | $\delta < 1.25^2$ ↑ | $\delta < 1.25^3$ ↑ |
|---|---|---|---|---|---|---|---|---|---|
| SelfOcc* [22] | ✔ | ✗ | 0.257 | 3.391 | 9.188 | 0.383 | 0.6379 | 0.8198 | 0.9022 |
| SelfOcc [22] | ✔ | ✔ | 0.348 | 5.554 | 10.556 | 0.442 | 0.611 | 0.775 | 0.863 |
| UniPAD [21] | ✔ | ✔ | 0.276 | 3.119 | 6.267 | 0.327 | 0.649 | 0.870 | 0.941 |

| Depth Distill | Virtual Distill | | | DistillNeRF Variants | | | | | |
|---|---|---|---|---|---|---|---|---|---|
| ✗ | ✗ | ✔ | ✔ | 0.270 | 3.670 | 6.301 | 0.389 | 0.653 | 0.826 | 0.886 |
| ✔ | ✗ | ✔ | ✔ | 0.235 | 3.008 | 5.859 | 0.311 | **0.726** | **0.890** | 0.942 |
| ✔ | ✔ | ✔ | ✔ | **0.228** | **1.898** | **5.654** | **0.302** | 0.689 | 0.879 | **0.943** |

## 4.1 Rendering Images and Foundation Features

**Setup.** We evaluate our model on previously unseen scenes from the validation set. For scene reconstruction, we compare the rendered images against GT images for the same time step. For novel-view synthesis, we render the novel-view image from the next timestep's camera pose, and compare it against the next timestep's GT image. We use standard metrics: peak signal-to-noise ratio (PSNR) and structural similarity index (SSIM). We compare with two SOTA generalizable NeRF methods in driving scenes, SelfOcc [22] and UniPAD [21], which do not see the validation set during training, similar to ours. We also compare with the SOTA per-scene optimized method, EmerNeRFs [14], which is trained on the validation set. Since EmerNerfs are trained on all timesteps in the scene, we cannot evaluate them for novel views. Instead, for novel-view evaluation we adapt *Single-Frame EmerNeRF*s, each of which is trained only on a single timestep and then evaluated for the next timestep. Due to the prohibitive training cost of Single-Frame EmerNerf on all timesteps, for all methods we report mean metrics over only the second frame of each scene.

**RGB Reconstruction and Novel-View Synthesis.** In Table 1, we show the results the image reconstruction and novel-view synthesis on the nuScenes validation set. The results show that our generalizable model is on par with the per-scene optimized NeRFs, and significantly outperforms SOTA generalizable methods, both for RGB reconstruction and novel-view synthesis. Without per-scene optimization, reconstruction PSNR for our best model variant is close to per-scene optimized EmerNerfs (30.11 vs 30.88), and achieves even higher SSIM (0.917 vs 0.879). Similarly, our novel-view PSNR is close to Single-Frame EmerNerf (20.78 vs 20.95), while SSIM is slightly higher (0.590 vs 0.585). Compared prior SOTA generalizable methods, our model outperforms the best-performing method (SelfOcc) in PSNR by 45.6% and 14.0%, and in SSIM by 64.9% and 27.1%, for reconstruction and novel-view synthesis, respectively. Novel-view metrics are generally lower than reconstruction metrics. Note that our novel-view setting is challenging, as the vehicle can travel large distances (in 0.5s) between the input-view and novel-view camera poses, and capture elements

that are invisible in the original camera pose. Further, dynamic objects that move between frames act as noise in our novel-view targets. Qualitative results are in Fig. 6. EmerNerf and our approach are close to the ground truth, while UniPAD generates blurry reconstructions with scan patterns, and SelfOcc generates grayish images and struggles to reconstruct the scene precisely.

**Generalization to Unseen Waymo NOTR Dataset** As in Fig 4 and Tab 3, we evaluate the generalizability of DistillNeRF to unseen domains. Trained on the nuScenes dataset, our model demonstrates strong zero-shot transfer performance on the unseen Waymo NOTR dataset. This quality can be further enhanced by applying simple color alterations to account for camera-specific coloring discrepancies. After fine-tuning, our model surpasses the offline per-scene optimized EmerNeRF on the data, achieving higher PSNR (29.84 vs. 28.87) and SSIM (0.911 vs. 0.814). We use the full Waymo NOTR data for evaluation, and quote original EmerNeRF metrics.

Table 3: Trained on the nuScenes dataset, Distill-NeRF shows strong generalizability to the unseen Waymo NOTR dataset.

| Method | PSNR | SSIM |
|---|---|---|
| EmerNeRF | 28.87 | 0.814 |
| Zero-Shot Transfer | 21.03 | 0.841 |
| Zero-Shot Transfer + Recolor | 24.85 | 0.867 |
| Finetune | **29.84** | **0.911** |

**Foundation Feature Reconstruction.** We choose the DistillNeRF variant with the best RGB reconstruction performance, and train it replacing the RGB image targets with feature image targets extracted from CLIP or DINOv2. Following EmerNerf, we reduce the dimensionality of target features to 64 dimensions using Principle Component Analysis (PCA). Results in Table 1 indicate that our method can successfully reconstruct CLIP and DINOv2 features, with a reconstruction performance not far from per-scene optimized EmerNerf. Note that EmerNerf additionally learns a separate positional-encoding head to denoise target features, which could also improve DistillNeRF results in the future. In Fig. 3 we show qualitative examples for foundation feature predictions, as well as results for utilizing the predicted features for open vocabulary scene understanding. Specifically, we obtain CLIP text embeddings for keywords, such "Car", "Building", "Road", and visualize the normalized similarity of the text embedding with rendered pixel-wise CLIP features. The results indicate the ability of DistillNeRF to understand rich semantics of the scene to a remarkable extent.

**Ablations.** Ablation results in Table 1 and Fig. 7 indicate that depth distillation from offline NeRFs increases reconstruction and novel-view synthesis performance, while virtual camera distillation benefits novel-view synthesis. The parameterized space slightly reduces the rendering metrics, but as shown in Fig 7, it is capable of generating unbounded depth.

## 4.2  Depth Estimation

**Setup.** We evaluate depth up to 80m using common metrics (Abs Rel, Sq Rel, RMSE, RMSE log, and $\delta < t$) [53, 54, 55]. We use two different depth targets: *Sparse LiDAR GT*, the common evaluation setting using LiDAR point cloud as ground truth, which is accurate but spare and has limited range (e.g. only 3m height); and *Dense Depth GT*, that uses EmerNerf to define dense depth targets with large range. We compare against the same baselines as for rendering. For UniPAD we increase the maximum range to 80m and retrain the model. For SelfOcc, we evaluate two model variants, *SelfOcc\** that supports depth prediction only (used in [22]), and *SelfOcc* that also supports rendering (thus more similar to our method). Same as before, we evaluate over the second frame of each scene.

**Depth Comparison.** Results in Table 2 and Fig. 6 show that while EmerNerf has superior depth accuracy by being optimized for each scene, our method outperforms prior SOTA generalizable NeRFs (SelfOcc and UniPAD). Specifically, while SelfOcc which only considers the depth prediction task shows high performance (noted as SelfOcc\*), when we evaluate the model that supports both depth and rendering (noted as SelfOcc), the performance drops considerably. Looking at Fig. 6, SelfOcc and UniPAD generate unreasonable depths for higher regions of the image, which is not reflected when evaluated against the sparse LiDAR ground truth. When evaluated on dense depth targets (Table 2b), their performance drops, while our approach shows more consistent performance for the two sources of ground truth.

**Ablations.** Consistently with previous results, distillation from offline NeRFs also improve depth estimation (Table 2). Quantitatively (Fig. 7), without depth distillation, we see inconsistent depth predictions between low and high regions of the image; without parameterized space, the model

Table 4: Unsupervised 3D occupancy prediction on the Occ3D-nuScenes [5] dataset. Our method learns meaningful geometry and reasonable semantics compared to alternative unsupervised methods. F-mIoU, mIoU and G-IoU denote the IoU for foreground-object classes, IoU for all classes, and geometric IoU ignoring the classes.

| Method | F-mIoU | mIoU | G-IoU | others | barrier | bicycle | bus | car | cons. veh. | motorcycle | pedestrian | traffic cone | trailer | truck | drive. surf. | other flat | sidewalk | terrain | mammade | vegetation |
|---|---|---|---|---|---|---|---|---|---|---|---|---|---|---|---|---|---|---|---|---|
| SimpleOcc [59] | 3.68 | - | 7.99 | - | 0.67 | 1.18 | 3.21 | 7.63 | 1.02 | 0.26 | 1.80 | 0.26 | 1.07 | 2.81 | 40.44 | - | 18.30 | 17.01 | 13.42 | 10.84 |
| OccNeRF [23] | 5.33 | - | 10.81 | - | 0.83 | 0.82 | 5.13 | 12.49 | 3.50 | 0.23 | 3.10 | 1.84 | 0.52 | 3.90 | 52.62 | - | 20.81 | 24.75 | 18.45 | 13.19 |
| SelfOcc (BEV) [22] | 2.71 | 6.76 | 44.33 | 0.00 | 0.00 | 0.00 | 0.00 | 9.82 | 0.00 | 0.00 | 0.00 | 0.00 | 0.00 | 6.97 | 47.03 | 0.00 | 18.75 | 16.58 | 11.93 | 3.81 |
| SelfOcc (TPV) [22] | 4.14 | 9.30 | 45.01 | 0.00 | 0.15 | 0.66 | 5.46 | 12.54 | 0.00 | 0.00 | 0.00 | 0.00 | 0.00 | 8.25 | 55.49 | 0.00 | 26.30 | 26.54 | 14.22 | 5.60 |
| Depth Distill | | | DistillNeRF Variants | | | | | | | | | | | | | | | | | |
| ✗ | 3.48 | 4.63 | 13.41 | 0.02 | 0.77 | 1.41 | 5.77 | 6.33 | 1.56 | 1.32 | 4.38 | 3.3 | 0.47 | 4.34 | 20.14 | 0.00 | 8.36 | 8.44 | 4.76 | 7.37 |
| ✔ | 6.40 | 8.93 | 29.11 | 0.03 | 1.35 | 2.08 | 10.21 | 10.09 | 2.56 | 1.98 | 5.54 | 4.62 | 1.43 | 7.90 | 43.02 | 0.00 | 16.86 | 15.02 | 14.06 | 15.06 |

can only predict depth in a limited depth range, while with parameterized space we can generate reasonable unbounded depth.

### 4.3 3D Semantic Occupancy Prediction

**Setup.** To evaluate the zero-shot downstream capabilities of DistillNeRF, we run evaluation on the Occ3D-nuScenes dataset [5] for 3d semantic occupancy prediction. The dataset comprises semantic occuapancy labels with 18 classes in the range [-40m, -40m, -1m, 40m, 40m, 5.4m] with voxel size 0.4m. We evaluate both binary and semantic 3d occupancy prediction. In DistillNeRF we use density thresholding (<0.001) to define whether a voxel is occupied. For semantic occupancy prediction, following [23], we use a pre-trained open vocabulary model GroundedSAM [56, 57, 58] to generate 2D semantic masks for the input images. Then, we project the center of occupied voxels onto the 2D masks to get the semantic class, following [22]. We found that the resolution of input and output images is important for occupancy prediction with DistillNeRF, so we increased them to 400×228 and 200×114, respectively.

We compare our method with SOTA self-supervised methods that do not use occupancy annotation: SimpleOcc [59], OccNeRF [23], and SelfOcc [22]. Following prior work, we report Intersection-over-Union (IoU) for each semantic category, mean IoU over all categories (mIoU), and geometry only IoU (G-IoU) for binary occupancy that ignores the semantic class. Additionally, we also report mean IoU over foreground categories (F-mIoU), that is, for categories excluding, *drivable surface*, *sidewalk*, *terrain*, *other flat*, and *others*.

**Results.** We show the comparison of occupancy prediction in Table 4. Our approach achieves competitive performance and excels on F-mIoU compared to the baselines, presumably because the sparse voxel representation emphasizes and better fits the foreground objects. SelfOcc (TPV) produces the highest mIoU and G-IoU, in part because it takes advantage of the fact that these metrics are dominated by ground-related classes (drive. surf., sidewalk, terrain), and it learns a prior for predicting the ground level as occupied even for non-visible regions (Fig.4 in [22]). Comparing ablations from DistillNeRF, we observe that distillation from offline NeRFs significantly improves performance (8.93 vs. 4.63 mIoU).

## 5 Conclusion

We proposed a framework for generalizable 3D scene representation prediction from sparse multi-view image inputs, using distillation from per-scene optimized NeRFs and visual foundation models. We also introduced a novel model architecture with spare hierarchical voxels. Our method achieved promising results in various downstream tasks.

Our approach is not without limitations. First, we currently rely on LiDAR to train offline EmerNerfs for distillation. Second, our sparse voxel representation naturally trades off rendering efficiency for dense scene representation, and thus may not be suitable for all downstream tasks. An interesting idea would be to combine a low-resolution dense voxel with a sparse voxel, or explore respresentations similar to Gaussian Splatting instead of voxels. Finally, there are numerous exciting directions for future work, including introducing temporal input, learning static-dynamic decomposition similar to EmerNerf, and utilizing the learned rich 3D scene representation for downstream tasks, such as detection, tracking, mapping, and planning.

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

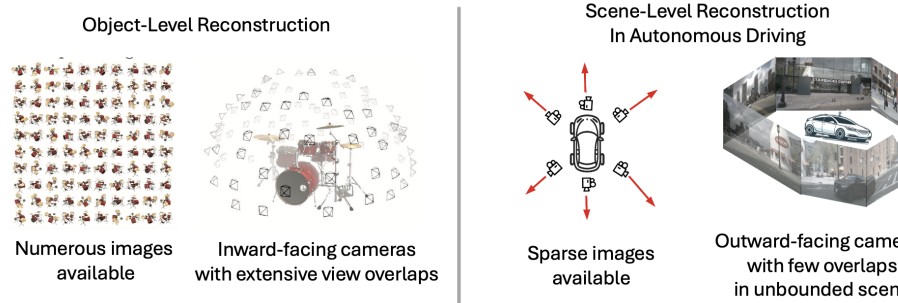

Figure 5: Scene-level reconstruction in autonomous driving poses different challenges compared to object-level reconstruction. 1) Typical object-level indoor NeRF involves an "inward" multi-view setup, where numerous cameras are positioned around the object from various angles. This setup creates extensive view overlap and simplifies geometry learning. In contrast, the outdoor driving task uses an "outward" sparse-view setup, with only 6 cameras facing different directions from the car. The limited overlap between cameras significantly aggravates the ambiguity in depth/geometry learning. 2) In the images captured from unbounded driving scenes, nearby objects occupy significantly more pixels than those far away, even if their physical sizes are identical. This introduces the difficulty in processing and coordinating distant/nearby objects.

## A  Appendix

### A.1  Discussion on Challenges in Driving Scenes

Compared to object-centric indoor scenes, the online autonomous driving with sparse camera images poses two key challenges, which is also illustrated in Fig 5:

1. *Sparse views with limited overlap complicate depth estimation and geometry learning*: Typical object-level indoor NeRF involves an "inward" multi-view setup, where numerous cameras are positioned around the object from various angles. This setup creates extensive view overlap and simplifies geometry learning. In contrast, the outdoor driving task uses an "outward" sparse-view setup, with only 6 cameras facing different directions from the car. The limited overlap between cameras increases the ambiguity in depth/geometry learning. To this end, following designs are made:

- Depth distillation: depth images rendered from per-scene NeRF are used to supervise our model. These per-scene optimized depth images are high-quality, dense, and consistent spatially/temporally
- Virtual camera distillation: virtual cameras are created as additional targets to artificially increase view overlaps
- Two-stage Lift-Splat-Shoot strategy: proposed to capture more nuanced depth
- Features from 2D pre-trained encoder: help the model learn better depth estimations and 2D image features

2. *Difficulty in processing and coordinating distant/nearby objects in the unbounded scene*: Unlike object-centric indoor problems, in the unbounded driving scene, the images usually contain unevenly distributed visual information based on distance: nearby objects occupy significantly more pixels than those far away, even if their physical sizes are identical. This is usually not the case in common NeRF benchmarks, and motivates multiple key designs:

- Parameterized space: introduced to account for the unbounded scene, unlike Self-Occ/UniPAD with a limited range ( 50m) and losing geometry information for far-away scenes.
- Density complement: far-away objects occupy pixels, and thus sampled rays could easily miss them (e.g. distant cars). Thus we propose to query densities from both fine/coarse voxel, and complement density.
- Light-weight upsampling decoder: applied to rendered feature images, to 1) upsample the final RGB image without additional rendering cost; 2) enable robustness to noises in rendered features

Table 5: Ablation studies on key components in our model.

| Density Complement | Decoder | Pretrained Encoder | Two-stage LSS | Two-Depth Distillation | PSNR | SSIM |
|---|---|---|---|---|---|---|
| ✗ | ✓ | ✓ | ✓ | ✓ | 22.76 | 0.669 |
| ✓ | ✗ | ✓ | ✓ | ✓ | 25.34 | 0.839 |
| ✓ | ✓ | ✗ | ✓ | ✓ | 21.35 | 0.536 |
| ✓ | ✓ | ✓ | ✗ | ✓ | 27.40 | 0.859 |
| ✓ | ✓ | ✓ | ✓ | ✗ | 28.01 | 0.872 |
| ✓ | ✓ | ✓ | ✓ | ✓ | 30.11 | 0.917 |

## A.2 Ablation Studies on Model Component

Following the discussions above, as shown in Tab 5, we ablated key components of our sparse voxel representation such as density complement, decoder, pre-trained 2D encoder, and the two-stage LSS. We remove one component each time to ablate its effect. In the last row, we also further add the depth distillation from offline NeRF, which represents the best performance of our full model.

- *No density complement*: we observe a significant drop of PSNR from 28.01 to 22.76, demonstrating the importance of better coordination between low-level and high-level sparse voxels.

- *No decoder*: we see a decent drop of PSNR from 28.01 to 25.76, showing the effectiveness of using the decoder for robustness to noises in rendered features.

- *No pre-trained 2D encoder*: we see a significant drop of PSNR from 28.01 to 21.35, which is expected since using pre-trained 2D encoder has been a commonly acknowledged approach in the field, which SOTA methods such as UniPAD and SelfOcc all adopt.

- *No two-stage LSS*: we observe a slight drop of PSNR from 28.01 to 27.40.

- *Add depth distillation from offline NeRF*: we observe the jump of PSNR from 28.01 to 30.11.

Note that every ablation above outperforms the SOTA methods UniPAD and SelfOcc, which have PSNR of 19.44 and 20.67 respectively.

## A.3 Discussion on Advantages of Different Ways of Lifting 2D to 3D.

In this work, we employ lift-splat-shoot with convolutions to lift 2D features to 3D. In comparison, another common approach of lifting is 3D-to-2D feature projection, which is exploited in two threads of generalizable approaches: 1) *image-based methods*, such as GeoNeRF, IBRNet; 2) *voxel-based methods*, such as UniPAD, NeRF-Det.

In the *image-based generalizable methods*, for one query point along the ray from novel views, features are extracted from the feature volume of surrounding source views via projection and interpolation. Our multi-view fusion/voxel grid approach possessed multiple advantages:

- Explicit Representation: Voxel-based methods provide a direct/explicit 3D scene representation, allowing more straightforward manipulation, analysis, and understanding of spatial relationships in the scene (e.g. removing or replacing object for simulation use)

- Scalability: As in our case, voxel-based methods can scale to different levels of detail by adjusting the voxel resolution. This scalability allows for efficient representation and rendering of both large scenes and fine details.

In *voxel-based generalizable works*, a 3D voxel is created, and the feature of each voxel is generated by projecting the voxel coordinates to 2D image features and feature interpolation, where an explicit representation is offered. However, the issues with this approach include

- As pointed out by NeRF-Det, this approach is equivalent to shooting a ray from the camera into the scene, and populating all voxels on the ray with the same 2D features, which could have strong depth ambiguity along the ray since all voxels on the ray have the same feature.

- Besides, such 3D-to-2D projection introduces non-trivial challenges for 3D voxel grids sparsification, since every voxel is populated with features.

Therefore, in this work, we choose to use the multi-view fusion approach and a lift-splat-shoot approach where

- the model predicts the depth of each pixel, and only lifts 2D features to voxels around the depth of the pixel;
- the model enables easy voxel sparsifications, which accelerates 3D encoding via sparse convolutions.
- depth frustums are explicitly created for each view, and the distillation from Per-Scene NeRF depth would be more straightforward (applying depth loss or regularization on the frustums).

As in our experiments, when comparing against UniPAD which takes the 3D-to-2D feature projection approach, our method shows better view synthesis and depth prediction performance, and offers higher inference efficiency. We hope to present our approach as an interesting and promising method that could be valuable to the community.

## A.4    Discussions and Ablations on Foundation Model Feature Distillation

In our paper, we distill the foundation model features into our model for enhanced semantics. The resulting model is able to create a 3D representation populated with foundation model features, and render 2D foundation model features from the 3D scene. However, to generate 2D foundation model feature images, there could be simpler baselines, e.g. rendering RGB and then feeding the RGB renderings into CLIP/DINO models. As shown in Tab 6, we compared the reconstruction accuracy and inference speed for the two approaches.

As expected, given the high-quality rendered RGB images from our model, directly feeding these rendered RGB images into CLIP/DINO models generates a good original-view reconstruction accuracy (CLIP: 19.81 vs. 18.69, DINO: 21.70 vs. 18.48). However, this baseline introduces significantly higher inference latency and memory consumption due to the additional use of CLIP/DINO models. Specifically, for the CLIP model, the inference time is three times longer (1.656s vs. 0.501s, 3.3 times), and for the DINO model, the inference time is almost twice as long (0.948s vs. 0.501s, 1.89 time).

Besides, we would like to emphasize that by distilling 2D foundation model features into our model, we not only enable it to render 2D foundation model feature images, but also lift 2D foundation models into 3D simultaneously. The resulting 3D voxel fields, similar to those in EmerNeRF [14] and LeRF [42], contain rich semantic information. As demonstrated by prior works such as LeRF-ToGo [60], ConceptFusion [61] and FeatureNeRF [20], such 3D foundational features can greatly facilitate 3D multimodal grounding (e.g. bridging language via CLIP features) and effectively benefit downstream tasks such as segmentation, keypoint transfer, and robot planning in open world.

Table 6: The reconstruction accuracy and inference speed of two approaches to generate foundation model feature images.

|  | CLIP Inference time (s) | CLIP PSNR | DINO Inference time (s) | DINO PSNR |
|---|---|---|---|---|
| DistillNeRF rendering RGB + FM model | 1.65651 | 19.81 | 0.94878 | 21.70 |
| DistillNeRF rendering FM (Ours) | 0.50175 | 18.69 | 0.50175 | 18.48 |

## A.5    Details on Training Losses

At the end of Sec 3, we introduced the training loss of our method:

$$L = \underbrace{L_{rgb} + L_{depth} + L_{density}}_{\text{rendering}} + \underbrace{L_{NeRF} + L_{found}}_{\text{distillation}}, \tag{5}$$

where $L_{rgb}$ and $L_{depth}$ are rendering losses for RGB and depth; $L_{density}$ is a density entropy loss as in [14]; $L_{NeRF}$ and $L_{found}$ denote distillation losses from offline NeRFs and foundation models. Here we introduce more details on each loss terms by describing their equations or pseudo code:

- $L_{rgb}$ comprises $L_1$ loss and perceptual loss:

$$||GT_{RGB} - Pred_{RGB}||_2 + LPIPS(GT_{RGB}, Pred_{RGB}) \qquad (6)$$

- $L_{depth}$ includes $L_1$ and MSE loss:

$$\frac{||GT_{depth} - Pred_{depth}||}{max\_gt\_depth} + \frac{||GT_{depth} - Pred_{depth}||_2}{max\_gt\_depth} \qquad (7)$$

where we normalize the absolute depth values to 0 1, and the $L_1$ and MSE losses are computed between ground truth depths (LiDAR points projected onto image planes) and predicted depths.

- $L_{density}$ is a density entropy loss from EmerNeRF, which encourages the opacity of a ray to be 1:

$$BEC(opacity, ones\_like(opacity)) \qquad (8)$$

, where opacity is the accumulated opacity per ray, and BCE is the binary cross entropy loss.

- $L_{nerf}$ captures the distillation from per-scene NeRFs, including: 1) Dense depth distillation loss: The same depth loss equation as above, but computed with a different ground-truth, namely dense depth maps rendered from per-scene NeRFs in original camera views. 2) Novel camera view supervision: the same RGB and depth losses above, but between the per-scene NeRFs' rendered results and our online models' predictions.

- $L_{found}$: $L_1$ loss on the predicted foundation model features:

$$||GT_{feats} - Pred_{feats}|| \qquad (9)$$

## A.6 Inference Time Analysis

In our paper, we demonstrate that our method substantially outperforms other SOTA generalizable approaches, specifically SelfOcc and UniPad, with a reconstruction PSNR of 30.11 compared to 20.67 and 19.44, respectively. To provide a thorough evaluation, we also analyzed the inference speed of our method alongside SelfOcc and UniPad, and present the results in Tab. A.6. The table details the inference time for each component in our method: our model requires a total of 0.4867s for inference, with 0.3594s dedicated to voxel prediction from six image inputs and 0.1273s for rendering six images (achieving approximately 47 fps for a single camera). In comparison, SelfOcc and UniPad achieve total inference times of 0.1771s and 0.6514s, respectively.

- As expected, SelfOcc achieves high inference speed, since it adopts an implicit representation where deformable cross-attention is used to aggregate information from the image features to generate a 3D SDF field. In comparison, our explicit voxel representation takes decent time for the 3D convolution operations, but offers additional benefits such as more straightforward manipulation (e.g. removing or replacing object for simulation use), and flexibility to scale to different levels of detail by adjusting the voxel resolution.

- UniPad utilizes a voxel-based representation, making it more directly comparable to our approach. Our method shows faster inference speed, presumably due to key designs such as voxel sparsification, and the lightweight decoder that enables efficient rendering.

Evaluations were conducted on the same desktop-grade system (13th Gen Intel(R) Core(TM) i7-13700KF, NVIDIA GeForce RTX 4090) and used the same image resolution ($228 \times 128$). To ensure accurate inference time measurement, we performed CUDA synchronization to exclude the time associated with tensor transfers.

## A.7 Implementation Details

The resolution of input RGB images, rendered RGB, and rendered depth are 114×228, 114×228, and 64×114 respectively. To favor downstream tasks (e.g. occupancy prediction typically considers the region of interest as [-40, -40, -1.6, 40, 40, 4.8]), in our parameterized neural field, the range of the inner voxel is 50 meters for the two horizontal directions and 6.4 meters for the vertical directions, and the proportion of the inner range $\alpha$ is set as 0.8. For the sparse voxel representation, we create two multi-resolution octrees with the finest levels of 7 and 9, respectively, where we employ the Kaolin library for implementation, ensuring robust and efficient handling of voxel data.

Table 7: Inference time comparison with SOTA methods, and a breakdown on each component in our model.

| Method (Component) | Run Time (s) | Reconstruction PSNR |
|---|---|---|
| SelfOcc | 0.1771 | 20.67 |
| UniPAD | 0.6514 | 19.44 |
| DistillNeRF (Ours) | 0.4867 | 30.11 |
| Encoder | 0.3594 | - |
| Single-view encoding | 0.0407 | - |
| Multi-view fusion | 0.3186 | - |
| Voxel convolution | 0.2494 | - |
| Renderer | 0.1273 | - |
| Projection + Ray march | 0.1264 | - |
| Decoder | 0.0008 | - |

During single-view encoding, to generate the depth feature map in the first stage, we feed each image to feature pyramid network (FPN) [62] to generate multi-scale fused features, which are then concatenated with prior depth features from [63] as the final depth feature map for the image. Specifically, we extracted the feature before the output layer as the prior depth feature. To generate the density of each depth candidate in the second stage, we first embed the depth candidate, then concatenate it with the depth feature map from the first stage, and further embed the concatenated feature. To generate the 2D image features, we use the same strategy as in the first-stage depth prediction, namely feeding each image to feature pyramid network (FPN) [62] to generate multi-scale fused features, which are then concatenated with prior depth features from [63].

Regarding the virtual camera distillation, for each of the six cameras, we move the camera pose 1 meter away from the original camera pose in three directions (upward, leftward, and rightward), to render virtual depths/RGB images from offline NeRFs. During training, for each camera, we randomly sample one virtual view for supervision in addition to the original camera view. To facilitate CLIP/DINOv2 feature synthesis, we use the PCA matrix to reduce the feature dimension from 768 to 64. The PCA matrix is generated according to random samples from ground-truth feature images, and is applied to all data samples.

We trained DistillNeRFs on 8x A100 GPUs, each with 80 GB memory, for around 4 days. We use a learning rate of 0.0002, and the Adam optimizer with an exponential decay rate of the moving averages $\beta_1 = 0, \beta_2 = 0.99$. We apply a gradient clip with a maximum 35 $l_2$ norm to stabilize training. At inference, our model takes around 1.708s for single-view encoding/lifting, and multi-view fusion, and around 0.685s for rendering RGB images, tested on a local machine with an NVIDIA TITAN RTX GPU, and an Intel(R) Core(TM) i9-10980XE CPU. Moderate training instabilities and oscillations were observed, particularly in depth prediction, which tended to fluctuate across epochs and runs. This behavior is likely due to the trade-off between geometry and rendering introduced by the sparse hierarchical design.

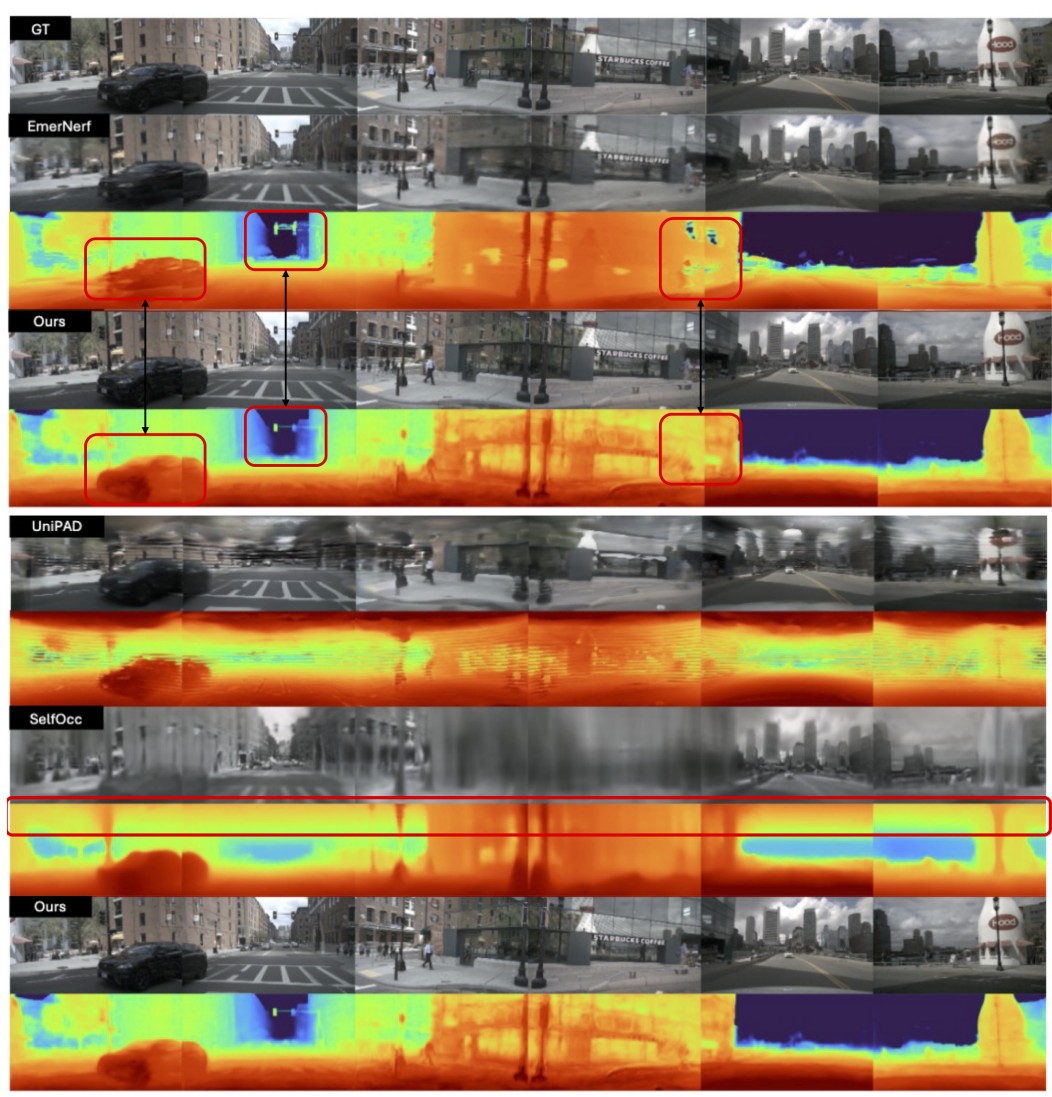

Figure 6: Qualitative comparison on RGB image and depth image reconstruction. Our generalizable DistillNerf is on par with SOTA offline per-scene optimized NeRF method (EmerNerf), and significantly outperforms SOTA generalizable methods (UniPAD and SelfOcc).

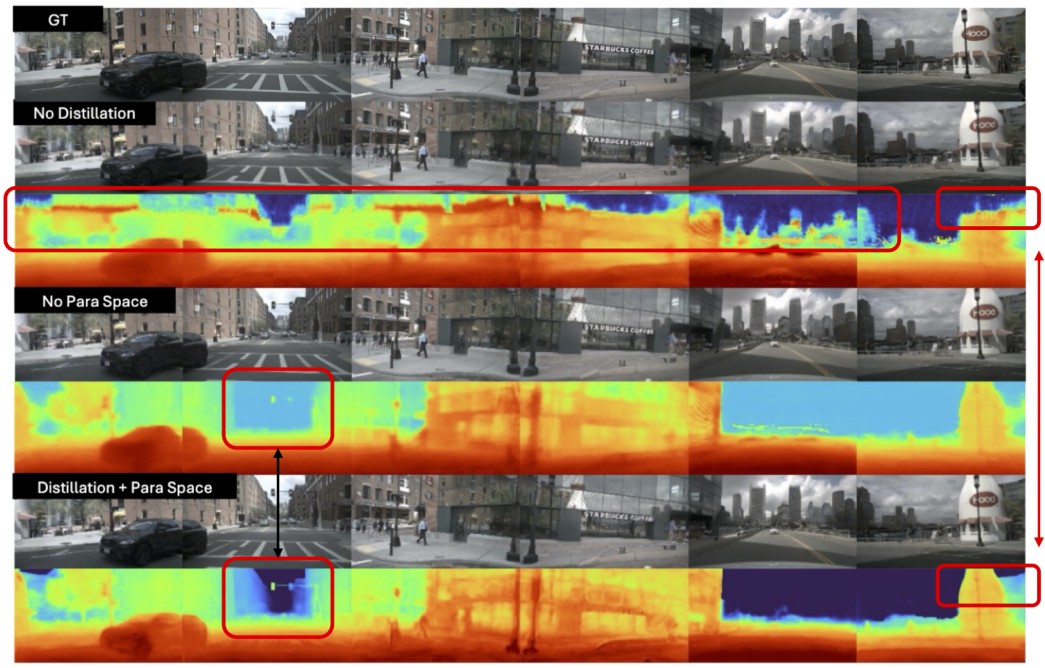

Figure 7: Qualitative ablation studies of our model on the RGB image and depth image reconstruction. Without depth distillation, we see inconsistent depth predictions between low and high regions of the image. Without parameterized space, the model can only predict depth in a limited depth range, while with parameterized space we can generate reasonable unbounded depth.

