# OpenReview forum: "DistillNeRF: Perceiving 3D Scenes from Single-Glance Images by Distilling Neural Fields and Foundation Model Features"
_NeurIPS.cc/2024/Conference — NeurIPS 2024 poster_

### Official Review · Reviewer_aV9Y · 2024-07-04

**Soundness:** 3
**Presentation:** 3
**Contribution:** 2
**Rating:** 5
**Confidence:** 3

**Summary:**

The paper introduces a self-supervised learning framework named DistillNeRF for understanding 3D environments from limited 2D observations. This framework is designed for autonomous driving and leverages per-scene optimized Neural Radiance Fields (NeRFs) and features distilled from pre-trained 2D foundation models such as CLIP and DINO. The model predicts rich 3D feature volumes from single-frame multi-view camera inputs, supporting various downstream tasks like scene reconstruction, novel view synthesis, and zero-shot 3D semantic occupancy prediction. Experimental results on the NuScenes dataset demonstrate the effectiveness of DistillNeRF over existing methods.

**Strengths:**

1. The paper is interesting to read and simple to follow. The methodology is robust, combining offline per-scene NeRF training with a distillation stage that generalizes across scenes.
2. The paper is well-structured and clearly explains the methodology, including detailed descriptions of the model architecture, training process, and experiments. Figures and tables effectively illustrate the key concepts and results.

**Weaknesses:**

1. Computational Complexity and Inference Speed: It would be beneficial for the authors to include the training time for both per-scene EmerNeRF and DistillNeRF, as well as the inference speed of DistillNeRF. This will help to better assess the efficiency of the proposed method.
2. Results on Other Datasets: Currently, the experiments are conducted only on the nuScenes dataset, whereas EmerNeRF has been tested on the Waymo dataset as well. Including experiments on the Waymo dataset would strengthen the claims regarding the generalizability of the method.

**Questions:**

See the "Weakness" section.
1. Since the nerf-based method requires extra training time, is it possible to extend this method to support 3DGS distillation?

**Limitations:**

See the "Weaknesses" section. Additionally, please note that the final score for this study is not solely determined by the peer reviewers' discussion. If the authors can address my main concerns, I would be willing to raise the score.

---

> ### Author Rebuttal · Authors · 2024-08-07
>
> Thank you for your positive feedback and valuable constructive comments! We address your specific questions below and will incorporate your suggestions into the paper. We have also included our source code during rebuttal (see the joint response), to enhance reproducibility and allow for inspecting every detail of our model.
>
> > Q1: Computational Complexity and Inference Speed
>
> Thank you for the suggestion to clarify the details of our approach!
> - As for training durations, EmerNeRF requires 1.5 to 2.5 hours per scene using a single A100 GPU (with flow field enabled). Given that NuScenes consists of 850 scenes, training EmerNeRF on the entire dataset would take approximately 1700 A100 GPU hours. Additionally, training our DistillNeRF model takes approximately 4 days using 8x A100 GPUs (as mentioned in Appendix L526), amounting to around 768 A100 GPU hours.
> - For detailed inference time breakdowns of our DistillNeRF, please refer to our joint response.
>
> > Q2. Results on additional dataset (Waymo)
> - This is a great suggestion. We initially chose the NuScenes dataset due to its popularity and because most state-of-the-art methods we compare against are studied solely on NuScenes. Also note that, from our experience, the NuScenes dataset presents more significant challenges compared to Waymo, such as using more cameras (6 vs. 3), lacking time synchronization for “sweep” frames, and having larger calibration errors and noise. A qualitative demonstration for NuScenes’ bigger challenge is that, EmerNeRF performs much worse in NuScenes (see Figure 4 in our paper) than Waymo (see the official EmerNeRF demo). Thus the evaluation on NuScenes could serve as reasonably convincing evidence.
> - While we do agree with the reviewer that it would be beneficial to include results from other datasets, limited time and compute constraints during the rebuttal period prevented us from finishing the training/evaluation in time. So far, we have completed the training of per-scene EmerNeRF and extracted the rendered depth image on the Waymo dataset, and will include these results in the final paper.
>
> > Q3. Extension to distillation from 3DGS
> - It’s a good point! While we initially chose EmerNeRF because its self-supervision nature aligns with our motivation, it would be generally straightforward to use any alternative method, since we just need to extract 2D depth/images from the offline method. If the alternative method can decompose dynamic and static objects, both depth distillation and virtual camera distillation are applicable. If not, depth distillation remains applicable.
> - 3DGS is a suitable choice due to its typically accelerated training and inference times, thanks to its efficient 3D projection, which would enhance the scalability of our pipeline.
>
> We are happy to engage in more in-depth discussions! Feel free to drop any questions or concerns you might have.

---

> > ### Comment · Reviewer_aV9Y · 2024-08-10
> >
> > Thanks to the author for providing the rebuttal. However, I agree with Reviewer WxYD that this work needs to be improved by including the materials in the rebuttal. Considering this point, I tend to lower my rating.

---

> ### Author Response · Authors · 2024-08-11
> **Response to reviewer's comment**
>
> Dear reviewer, first we would like to extend our sincere gratitude for taking the time to read through the feedback and engage in the discussion for our paper. We greatly value your precious time/efforts that you've dedicated to contributing to the conference and the broader community.
>
> We would like to clarify the point mentioned by R-WxYD, who stated that “this feels like writing the paper after the publication deadline”. However, as mentioned in our new response to R-WxYD, we are currently at the rebuttal stage rather than the publishing stage. We firmly believe, “reviewers provide constructive comments and authors improve the paper accordingly”, is one key reason to set up the rebuttal stage, and one important part of the review cycle where reviewers and authors work together to present greater works and contributions to the community. We kindly refer you to the NeurIPS reviewer guidelines, which emphasize the importance of this stage.
>
> Since you are taking reviewer WxYD's comments into your assessment, we would also gently suggest you read our responses to R-WxYD, as it could provide additional contexts that could be helpful to your evaluation.
>
> Thank you again for your time and thoughtful consideration.

---

### Official Review · Reviewer_aj67 · 2024-07-08

**Soundness:** 3
**Presentation:** 3
**Contribution:** 3
**Rating:** 6
**Confidence:** 4

**Summary:**

This paper presents a method for 3d understanding from 2d observations for autonomous driving. The main technical contribution is a feed-forward model, which is trained by distilling RGB and depth from a per-scene optimized NeRF model. The proposed model predicts 3d feature volumes that enable volumetric rendering similar to NeRFs. Experimental results on NuScenes show improved performance compared to recent comparable generalizable methods.

**Strengths:**

- The model is generalizable, meaning that no per-scene optimization is required at inference. Compared to recent comparable generalizable methods [18,19], the results for reconstruction, novel view synthesis and depth estimation are improved.
- The authors propose a simple methodology to distill depth and RGB renderings from EmerNeRF, which requires per-scene optimization. The distillation process improves the results.
- The method is extensively evaluated on NuScene, and achieves very good results for multiple tasks, namely novel view synthesis, depth estimation and also 3d occupancy prediction.

**Weaknesses:**

- One of the main reasons for having this generalizable formulation instead of just using e.g. EmerNeRF is that it is faster, yet inference time is not discussed in the paper. Can the authors report this in the rebuttal? For Table 1, 2 and 3 inference time should be reported for all methods.
- The ablations are unclear. What exactly is meant by “Ours” in Table 1, 2 and 3? Is it the model without depth distillation, param space and visual cam distillation? Why is not the model that combines all components like “Ours (+ Depth Distillation + Param Space + Virtual Cam Distillation)” tested? Since “Ours” performs worse than “Ours (+ X)” it looks like it is not tested. Why not test “Ours (full)” and e.g. “Ours (- Depth Distillation)”, more like a standard ablation study?
- The part about distilling foundation models is incomplete. There are some simple baselines missing, e.g. rendering RGB and then computing CLIP/DINO from the RGB renderings. How much slower is it and how does the reconstruction accuracy compare? It is also unclear why the foundation model reconstruction is only reported for the model variant “Ours (+ Depth Distillation)” in Table 1 and not the others.

**Questions:**

Please see weaknesses

**Limitations:**

This is adequately addressed

---

> ### Author Rebuttal · Authors · 2024-08-07
>
> Thank you for the positive feedback and the insightful comments! Our detailed response for each question is below. Note that we have also included our source code during rebuttal (see the joint response), to enhance the reproducibility and allow for inspecting every detail of our model.
>
> > Q1. Discussion on the inference time
>
> Please refer to the joint response for the detailed breakdowns of the inference time. We’re happy to provide more information if needed.
>
> > Q2. The ablations are unclear.
>
> Thank you for bringing this to our attention. To clarify, we conducted a standard ablation study where each row in Table 1 progressively adds a component to the model. Specifically, the entries in Table 1 correspond to the following configurations:
>
> |                                       |      Depth Distillation      |      Param Space      |      Virtual Cam Distillation      |
> |:-------------------------------------:|:----------------------------:|:---------------------:|:----------------------------------:|
> |                Ours                   |              ✗               |           ✗           |                  ✗                 |
> |    Ours (+ Depth Distillation)        |              ✓               |           ✗           |                  ✗                 |
> |           Ours (+ Param Space)        |              ✓               |           ✓           |                  ✗                 |
> | Ours (+ Virtual Cam Distillation)     |              ✓               |           ✓           |                  ✓                 |
>
> We recognize that the naming convention may have caused misunderstandings and we will update the paper to explicitly state these configurations and ensure clarity.
>
> > Q3: The part about distilling foundation models is incomplete, one ablation is required
>
> Thank you for the insightful comments. We have now added the ablation of rendering RGB images and then computing CLIP/DINO from these renderings, and compared the accuracy and inference speed. Our findings are reported below, where the inference time of the new baseline includes the forward pass of our RGB rendering model (0.48672s) and the CLIP/DINO model.
>
> |                                     | CLIP Inference time (s)     | CLIP PSNR | DINO Inference time (s)     | DINO PSNR |
> |:-----------------------------------:|:---------------------------:|:---------:|:---------------------------:|:---------:|
> | DistillNeRF rendering FM            | 0.50175                     | 18.69     | 0.50175                     | 18.48     |
> | DistillNeRF rendering RGB + FM model| 1.65651                     | 19.81     | 0.94878                     | 21.70     |
>
> As expected, given the high-quality rendered RGB images from our model, directly feeding these rendered RGB images into CLIP/DINO models generates a good original-view reconstruction accuracy (CLIP: 19.81 vs. 18.69, DINO: 21.70 vs. 18.48). However, this baseline introduces significantly higher inference latency and memory consumption due to the additional use of CLIP/DINO models. Specifically, for the CLIP model, the inference time is three times longer (1.656s vs. 0.501s, 3.3 times), and for the DINO model, the inference time is almost twice as long (0.948s vs. 0.501s, 1.89 time).
>
> Lastly, we would like to emphasize that by distilling 2D foundation model features into our model, we not only enable it to render 2D foundation model feature images, but also lift 2D foundation models into 3D at the same time. The resulting 3D voxel fields, similar to those in EmerNeRF and LeRF, contain rich semantic information. As demonstrated by prior works such as LeRF-ToGo [1], ConceptFusion[2] and FeatureNeRF[3], such 3D foundational features can greatly facilitate 3D multimodal grounding (e.g. bridging language via CLIP features) and effectively benefit downstream tasks such as segmentation, keypoint transfer, and robot planning in open world.
>
> [1] Rashid, Adam, et al. "Language embedded radiance fields for zero-shot task-oriented grasping." 7th Annual Conference on Robot Learning. 2023.
>
> [2] Jatavallabhula, Krishna Murthy, et al. "Conceptfusion: Open-set multimodal 3d mapping." Robotics: Science and Systems. 2023
>
> [3] Ye, Jianglong, Naiyan Wang, and Xiaolong Wang. "Featurenerf: Learning generalizable nerfs by distilling foundation models." Proceedings of the IEEE/CVF International Conference on Computer Vision. 2023.

---

> > ### Comment · Reviewer_aj67 · 2024-08-13
> >
> > I thank the authors for providing answers to all my questions. I keep my rating as weak accept.

---

### Official Review · Reviewer_WxYD · 2024-07-10

**Soundness:** 1
**Presentation:** 1
**Contribution:** 1
**Rating:** 3
**Confidence:** 3

**Summary:**

Appears to present a method for online 2D feature distillation in an autonomous driving configuration. Method appears to use some kind of pre-trained depth prediction network to build a frustum aligned grid of 2D features, which are somehow rasterized into a canonical sparse volumetric grid. There appears to be some use of networks ("embedding") to do inference on these features, which allow for 3D semantic segmentation and occupancy prediction.

Method presents novel view synthesis, depth prediction and occupancy prediction metrics on the nuScenes dataset.

**Strengths:**

Paper claims to be the first to achieve online distillation of 2D features into 3D in the autonomous driving domain. Presents some SotA metrics on novel view synthesis

**Weaknesses:**

The papers greatest weakeness is the lack of enough detail descrbing the method to reproduce it, and therefore to be able to properly critique it.  Distilling 2D features into 3D via Neural Radiance Fields is not a new idea (“LERF” , and “FeatureNeRF" are cited.  “Decomposing NeRF for Editing via Feature Field Distillation”, NeurIPS 2020 is not cited), yet is given too much attention. The real contribution should be as to how this is done in real-time, but critical details as to how 2D features are lifted into 3D via depth predictions, are missing, how view frustums are combined into a canonnical 3D grid are missing, and how features are combined and "embedded" (suggesting the use of a neural network) missing. (See "Questions below for further deails"). I would have thought the use of sparse grids would more easily allow unbounded spatial discretization and the MipNeRF-360 style space contraction unnecessary ("Neural Field Parameterization"). This design choice appears to be unmotivated and un-ablated.

Precise (non-verbal) definitions of all terms in the final loss Eq (4). appear to be missing.

Method is presented as an online method, but provides no metrics indicating real-time performance.

**Questions:**

#124 “depth feature map is further embedded”  - how do you perform this embedding? This would suggest you are using a neural network of some sort.

#127 “To this end, we first aggregate the frustum to predict a raw depth for each pixel, and then sample fine-grained depth candidates centered around the raw depth. Both depths are trained end-to-end with our model.” You provide equations for compositing (1) and expected depth (2) (which are relatively trivial and could probably be omitted), but exactly how you do the remainder appears to be missing.

#130 “the frustum is designed to contain the density value”. This would be the place to describe this design.

#136 “The depth feature map” It is unclear what you mean by “depth features”. Are intermediate features used to predict depth?  Bucketized depth probabilities? This might reference details that are buried inside “Depth Anything”, but not apparent on a cursory read of it.

#145 Unclear what you mean by “sparse quantization”. Quantization usually means discretizing the floating point value to a finite set of values. Is this what you mean? Also, what spatial interpolation and sampling technique are you using to sample from the frustum to the global grid?

#151 What do you mean by “two octrees with different quantization granularities,” An octree is by definition multi-resolution. Are you simply using a sparse grid? Do you have two sparse grids? If using an octree, please describe at least how many levels are in it. Further saying you apply “sparse convolutions” makes me think you actually mean sparse grid, because a sparse multi-resolution convolution on an octree is exceedingly difficult to implement.

**Limitations:**

Briefly mentions limitations in the final paragraph of "Conclusions". Though they state #334 "sparse voxel representation naturally trades off rendering efficiency for dense scene representation", no actual performance metrics appear to be given.

Societal impact is properly addressed.

---

> ### Author Rebuttal · Authors · 2024-08-07
>
> Thank you for your comments about clarity and reproducibility. To address your concerns, we have included the source code (see the joint responses), which allows for inspecting every detail of our model and reproducing results. We also address your specific questions below and will incorporate these details into the paper:
>
> >Q1:lack of enough details describing the method:
> - **Is distillation given too much attention?** We emphasize distillation as an important novelty because, unlike prior work, we not only distill foundation model features but also offline-trained NeRFs (depth distillation and virtual camera distillation). This approach is unique and has a significant impact on performance (see L298-302 and ablation studies in Tables 1, 2, and 3).
> - **Is space contraction necessary?** The use of sparse grids indeed allows us to process larger grids but it does not necessarily mean we can have an infinitely large grid. First, we rely on 3D convolution to propagate information among voxels that are populated by shooting rays from each pixel. As the rays travel farther away from cameras, the voxels the rays hit will be farther from each other, making convolution ineffective. Second, we estimate depth through occupancy predictions for a fixed number of entries in the view frustum, that would need to be fundamentally changed to allow unbounded predictions, which is not trivial.
> - **Definitions of loss terms in Eq (4)**: Eq.4 consists of 5 loss terms. Due to the character limit, we briefly introduce them here, and refer to the code for full details.
>     - **L_rgb** comprises L1 loss and perceptual loss: `rgb_loss=||GT_RGB-Pred_RGB||_2 + LPIPS(GT_RGB, Pred_RGB)`
>     - **L_depth** includes L1 and MSE loss: `depth_loss=||GT_depth - Pred_depth|| / max_gt_depth + ||GT_depth - Pred_depth||_2 / max_gt_depth`. Depth values are normalized to 0~1, and the L1 and MSE losses are computed between ground truth depths (from projecting lidar points onto image planes) and predicted depths.
>     - **L_density** is a density entropy loss from EmerNeRF, which encourages the opacity of a ray to be 1: `BEC(opacity, ones_like(opacity))`, where opacity is the accumulated opacity per ray, and `BCE` is the binary cross entropy loss.
>     - **L_nerf**: Involves distillation from per-scene NeRFs, including: 1) Dense depth distillation loss: The same depth loss but computed on dense depth maps rendered from per-scene NeRFs in original camera views. 2) Novel camera view supervision: Computes the same RGB and depth losses between the per-scene NeRFs' rendered results and our online models' predictions. We refer to the first term as dense 2D depth distillation loss and the second terms as virtual camera loss. We will add more details to existing descriptions (L188-209).
>     - **L_found**: L1 loss: `||GT_feats - Pred_feats||`
> - **Inference times**: please see our joint response, where we analyzed our inference time and compared it with other SOTA generalizable methods.
>
> >Q2. More Clarification on the Implementation Details
>
> 1.**“Depth feature map is further embedded”**:
>
> Indeed we used a neural network for the embedding operation, specifically 2D convolution layers. This helps in refining the depth feature map by leveraging spatial context.
>
> 2.**Details in the two-stage depth prediction strategy**:
>
> The coarse depth and fine depth undergo the same compositing and depth rendering process. We uniformly sample a fixed number of fine-grained depth candidates centered around the raw depth, where the coarse depth determines the sample range. This two-stage process allows for more precise depth estimation by first providing a coarse prediction and then refining it with finer details.
>
> 3.**“The frustum is designed to contain the density value”**:
>
> We model the depth as a set of predefined bins, which turns the depth regression problem into a depth classification problem, i.e., classifying the depth of a target pixel into a predefined depth bin. To implement this and remain simple, we use density value as the logits to compute the depth probability distribution via softmax operation (as mentioned by the reviewer, it’s “Bucketized depth probabilities”). This design choice ensures that the frustum effectively captures the spatial distribution of density values. In fact, many works are following this good practice, for example [1,2,3].
>
> 4.**The depth feature map**:
>
> Specifically, we extracted the feature from the pre-trained 2D encoder before the output layer. This approach captures rich semantic information from the 2D images, which is then used in our 3D model.
>
> 5.**What is sparse quantization**:
>
> Sparse quantization is the process of creating sparse voxels, such as in an octree structure, where only voxels with active inputs are created and stored. This contrasts with dense quantization, where every voxel is created and stored, even if no inputs or features are present in those voxels. Sparse quantization is a well-established technique in the literature, optimizing memory usage and computational efficiency.
>
> 6.**Two octrees with different quantization granularities**:
>
> We created two multi-resolution octrees with the finest levels of 7 and 9, respectively. This approach allows us to manage different levels of detail and spatial resolution effectively. We employed the Kaolin library to implement this sparse quantization, ensuring robust and efficient handling of voxel data.
>
> We believe these clarifications will enhance the understanding of our method and address the concerns raised. Thank you again for your valuable feedback. We are glad to provide more context and information if needed.
>
> [1] "Neuralfield-ldm: Scene generation with hierarchical latent diffusion models." CVPR 2023.
>
> [2] "Lift, splat, shoot: Encoding images from arbitrary camera rigs by implicitly unprojecting to 3d." ECCV 2020.
>
> [3] "Pixelsplat: 3d gaussian splats from image pairs for scalable generalizable 3d reconstruction." CVPR2024.

---

> > ### Comment · Reviewer_WxYD · 2024-08-09
> >
> > I thank the authors for their rebuttal.
> >
> > I remain very negative over the "lack of enough detail descrbing the method to reproduce it". I appreciate the authors inclusion of the source code, but there's no reason a well written Methods section _couldn't_ have included include all salient details so that I wouldn't _have to_ look at the source code. A paper should do the work of digesting the method for the reader instead of forcing them to go a level deeper and dissect the source code. Source code inclusion should only be for reproducibility purposes, and not as the primary documentation of the method. Furthermore, the authors rebuttal of "Due to the character limit, we briefly introduce them here, and refer to the code for full details." does not sound like you intend to include these losses in the main text.
> >
> > I appreciate the extra descriptions of the method. But these are all descriptions that should have been included in the paper. This feels like writing the paper after the publication deadline. As do the extra timing ablations that should have been included in the paper.
> >
> > I'm holding up the terminology of "sparse quantization" as unclear writing. I have always seen "quantization" refer to the quantization of floating point values, and not the sparsity structure of the grid. Can you provide a reference that actually uses this term the same way?
> >
> > I also have a hard time accepting 2fps at a very low resolution of 228x128 as a "realtime" or "online" method
> >
> > "... [we] ... also offline-trained NeRFs" I'm unclear what you are trying to say. Are you trying to say that you distilled features offline? Again, the language is still unclear.
> >
> > I appreciate the authors rebuttal but I maintain my rating as "reject".

---

> ### Author Response · Authors · 2024-08-11
> **Response to Reviewer WxYD**
>
> We thank you for the prompt response.
>
> > Q1: "lack of enough detail descrbing the method to reproduce it"...
> - First, to clarify, we firmly believe that our original submission has already described the method in sufficient detail, that is already up to the standards of the field. This is also reflected by the fact that, other reviewers unanimously rated the presentation as “3 good”, and explicitly mentioned “The paper is well-structured and clearly explains the methodology, including detailed descriptions of the model architecture, training process, and experiments” and “The proposed method exhibits good soundness, with the technical details of each component well elaborated”.
> - In such contexts, you raised questions such as the definitions of feature maps and embeddings. We firmly believe these are preliminary knowledge in deep learning/vision, but we respect/appreciate your unique angle, and were happy to respond to them one by one in the rebuttal phase.
> - Third, the reviewer suggested including every fine-to-the-ground detail of the paper, for which we even wrote down the mathematical equation of L1 loss. This is not actionable: today’s AI fields are moving very fast and have grown to a level of complexity that every paper would rely on previous foundations. It has been a common writing paradigm for papers to 1) mostly emphasize the core novelty of the paper, since that’s the key message delivered to the community; 2) leave the commonly used techniques or implementation details to references, appendix, supplementary material, and ultimately the source code. We provide code not as a replacement for a clear description, but as an additional resource to aid easy reproducibility, which is common practice in the field. For example, all SOTA methods compared in our paper follow this paradigm (EmerNeRF, UniPAD, SelfOcc, OccNeRF, SimpleOcc)
> - Based on the points above, we firmly believe that the reviewer’s concerns of clarification and reproducibility are already well addressed.
>
> > does not sound like you intend to include these losses in the main text.
> - As in our rebuttal, we explicitly said “will incorporate these details into the paper”. We kindly request you carefully read our response, and make informed comments.
>
> > I appreciate the extra descriptions of the method. But these are all descriptions that should have been included in the paper. This feels like writing the paper after the publication deadline. As do the extra timing ablations that should have been included in the paper.
> - Note that we are at the rebuttal stage instead of the publishing stage. We firmly believe, “reviewers provide constructive comments and authors improve the paper accordingly”, is one key reason to set up the rebuttal stage, and one important part of the process where reviewers and authors work together to present greater works and contributions to the community. Please also see the NeurIPS reviewer guidelines. It’s a pity to hear that you do not recognize the responses and improvements made during the rebuttal, even if they well address your concerns.
>
> > "... [we] ... also offline-trained NeRFs" I'm unclear what you are trying to say. Are you trying to say that you distilled features offline?
> - This is the key/core method of our paper, namely distillation from offline-trained NeRFs, which is clearly indicated by the title of our paper (DistillNeRF), and extensively elaborated/evaluated throughout the paper (Intro/abs, Fig.1, Sec 3.2, Table 1/2/3). Again, we kindly request you read our paper carefully and make informed comments.
>
> > I have always seen "quantization" refer to the quantization of floating point values, and not the sparsity structure of the grid. Can you provide a reference that actually uses this term the same way?
> - In 3D vision, voxel quantization, or sometimes called voxelization, is a basic operation widely used [1,2,3,4,5,6,7,8]. [1] introduced “Commonly, point clouds are first quantized in a process known as voxelization, with the resulting voxel grid being used as input to 3D CNNs”. Also, see Sec.3.1 in [2] for a whole section of descriptions. Fig.4 in [2] and Fig.1 in [3] also show excellent illustrations for dense and sparse quantization, respectively.
>
> > I also have a hard time accepting 2fps at a very low resolution of 228x128 as a "realtime" or "online" method
> - First, we use the term “online model” in line with the literature, referring to reconstructing the scene with one model forward pass [9,10,11]. This contrasts with “offline" NeRFs, which obtain a single scene representation through dedicated optimization. Our method renders 6 images in 0.486 seconds, compared to an offline NeRF like EmerNeRF, which takes 1.5–2.5 hours to achieve the same.
> - Second, we need to point out that our model generates 6 images with 0.486s, showing 12fps instead of 2fps. The pure rendering takes 0.127s with 47fps. Again, please read our response carefully and make informed comments.

---

> ### Author Response · Authors · 2024-08-11
> **Reference in our response**
>
> [1] Zhang, Chris, Wenjie Luo, and Raquel Urtasun. "Efficient convolutions for real-time semantic segmentation of 3d point clouds." 2018 International Conference on 3D Vision (3DV). IEEE, 2018.
>
> [2] Qian, R., Garg, D., Wang, Y., You, Y., Belongie, S., Hariharan, B., Campbell, M., Weinberger, K.Q. and Chao, W.L., 2020. End-to-end pseudo-lidar for image-based 3d object detection. In Proceedings of the IEEE/CVF conference on computer vision and pattern recognition (pp. 5881-5890).
>
> [3] Huang, Lila, et al. "Octsqueeze: Octree-structured entropy model for lidar compression." Proceedings of the IEEE/CVF conference on computer vision and pattern recognition. 2020.
>
> [4] Chen, Xiaozhi, et al. "Multi-view 3d object detection network for autonomous driving." Proceedings of the IEEE conference on Computer Vision and Pattern Recognition. 2017.
>
> [5] Ku, Jason, et al. "Joint 3d proposal generation and object detection from view aggregation." 2018 IEEE/RSJ International Conference on Intelligent Robots and Systems (IROS). IEEE, 2018.
>
> [6] Liang, Ming, et al. "Deep continuous fusion for multi-sensor 3d object detection." Proceedings of the European conference on computer vision (ECCV). 2018.
>
> [7] Yang, Bin, Wenjie Luo, and Raquel Urtasun. "Pixor: Real-time 3d object detection from point clouds." Proceedings of the IEEE conference on Computer Vision and Pattern Recognition. 2018.
>
> [8] Zhou, Yin, and Oncel Tuzel. "Voxelnet: End-to-end learning for point cloud based 3d object detection." Proceedings of the IEEE conference on computer vision and pattern recognition. 2018.
>
> [9] Yu, A., Ye, V., Tancik, M. and Kanazawa, A., 2021. pixelnerf: Neural radiance fields from one or few images. In Proceedings of the IEEE/CVF conference on computer vision and pattern recognition (pp. 4578-4587).
>
> [10] Charatan, D., Li, S.L., Tagliasacchi, A. and Sitzmann, V., 2024. pixelsplat: 3d gaussian splats from image pairs for scalable generalizable 3d reconstruction. In Proceedings of the IEEE/CVF Conference on Computer Vision and Pattern Recognition (pp. 19457-19467).
>
> [11] Wang, Q., Wang, Z., Genova, K., Srinivasan, P.P., Zhou, H., Barron, J.T., Martin-Brualla, R., Snavely, N. and Funkhouser, T., 2021. Ibrnet: Learning multi-view image-based rendering. In Proceedings of the IEEE/CVF conference on computer vision and pattern recognition (pp. 4690-4699).

---

> > ### Comment · Reviewer_WxYD · 2024-08-13
> >
> > I thank the authors for their response, but not their combative tone ("It’s a pity to hear that you do not...", "but we respect/appreciate your unique angle...", "We kindly request you carefully read our response, and make informed comments").
> >
> > There may be a cognitive bias to apply a perception of laziness and mis-understanding to the reader, but I would suggest that a more productive tone would be to ask if what your are intending to communicate is actually what you are communicating?
> >
> > As a meta-example, you state "[you] will incorporate these details into the paper". Yet closer to your response on loss terms you further state: "Due to the character limit, we briefly introduce them here, and refer to the code for full details." This reads as a) an excuse as to why these details weren't initially included, and b) an indication that the source code will be the documentation for these losses. There is no statement that you intend to include these loss terms in the paper. The wording "brielfy introduce them here" is ambiguous. What does "here" mean? The rebuttal? or the paper? A clearer response would have been: "We agree the loss equations are essential details originally omitted due to space constraints. We will amend our submission to include the terms listed below".
> >
> > Ultimately, you state "we firmly believe that our original submission has already described the method in sufficient detail, that is already up to the standards of the field." Which I disagree with (with all due respect to my peer reviewers who may feel the opposite). I do not hear the authors stating unambigiously they intend to clarify their presentation.

---

> ### Author Response · Authors · 2024-08-13
> **Response to reviewer's comment**
>
> Dear reviewer, thanks for the reply.
>
> We want to clarify, please do not get us wrong, our response is fully respectful to the reviewer and the reviewer's efforts, just like the word literally means "respect/appreciate", "kindly", and just as how we respond to reviewer's questions one by one in detail during the rebuttal stage, such as explaining the definition of feature maps, embeddings, voxel quantizations, write down the mathematical definition of l1 loss to address the reviewer’s question, and introduce the common practice for writing and code sharing in the community. Apologies if any of these comments look improper from some angles, that is not what we mean.
>
> We will incorporate the clarifications into our paper, just as we will incorporate every other reviewer's suggestions into our paper, and sincerely appreciate every reviewer's great suggestions/perspectives to improve our paper and contribution to the conference and community. Specifically, for the losses, we reassure the reviewer that we will update our paper to include them, just as how we responded to you in detail during our rebuttal.
>
> Finally, we thank the reviewer again for the precious time and effort in reviewing our work, and engaging in the discussion stage.

---

### Official Review · Reviewer_T1bA · 2024-07-31

**Soundness:** 3
**Presentation:** 3
**Contribution:** 3
**Rating:** 5
**Confidence:** 3

**Summary:**

This work aims to enhance the understanding of 3D environments from limited 2D observations in autonomous driving scenarios. It achieves this by proposing a new generalizable NeRF pipeline, trained using distillation from per-scene NeRFs and foundation models. This pipeline can transform input RGB images into 3D feature volumes that can be decoded into foundation model features at inference time in a feed-forward manner. Extensive experiments validate that the proposed method outperforms the baselines across various 3D tasks.

**Strengths:**

1. The proposed method exhibits good soundness, with the technical details of each component well elaborated.

2. The proposed method outperforms the baseline methods across different tasks and achieves results that are on par with per-scene optimization methods.

**Weaknesses:**

1. The new insights delivered by this work to the community are somewhat unclear. Specifically, this work combines various techniques, some of which have been partially employed in previous image-based rendering pipelines, into the proposed framework. However, it is unclear which techniques are particularly useful for the target autonomous driving scenario. If the authors aim to address the challenges specific to autonomous driving scenes, they should explicitly formulate these challenges and provide an analysis of which design is particularly useful for addressing each challenge. Although the detailed techniques may not be new, this analysis can significantly benefit the community when addressing similar scenes.

2. The differences between the components in the proposed pipeline and other image-based rendering pipelines need more clarification. Specifically, what are the key differences between the proposed method and the combination of GeoNeRF and FeatureNeRF? In my understanding, the key differences are the distillation from per-scene NeRF due to the lack of accurate depth and the adoption of multi-view fusion.

3. As a follow-up to point 2, I wonder what the advantages of adopting multi-view fusion plus a voxel grid are compared to 3D-to-2D feature projection for each sampled point along the ray in previous generalizable NeRFs.

4. Although this work mentions the real-time processing demand, the efficiency aspect of the proposed pipeline is not measured or analyzed.

5. I wonder whether the proposed method can be applied to common NeRF benchmarks, in addition to NuScenes, while still achieving leading generalizable reconstruction performance compared to other generalizable NeRF variants like IBRNet, GNT, and GeoNeRF.

**Questions:**

My questions have been included in the weakness section. I'm willing to adjust my scores if my concerns are properly addressed.

**Limitations:**

This work does not suffer from notable negative societal impacts.

---

> ### Author Rebuttal · Authors · 2024-08-07
>
> Thank you for the positive feedback and constructive comments! See the detailed response below, which will also be updated in our paper.
>
> > Q1: The new insights of this work are somewhat unclear. Formulate the challenges, and analyze which design is useful for them.
>
> Instead of object-centric indoor scenes, we target online autonomous driving with sparse images, which poses two key challenges: (also illustrated in Fig 2 of PDF attached)
>
> 1. _Sparse views with limited overlap complicate depth estimation and geometry learning_: Typical object-level indoor NeRF involves an "inward" multi-view setup, where numerous cameras are positioned around the object from various angles. This setup creates extensive view overlap and simplifies geometry learning. In contrast, the outdoor driving task uses an "outward" sparse-view setup, with only 6 cameras facing different directions from the car. The limited overlap between cameras increases the ambiguity in depth/geometry learning. To this end, following designs are made:
>
> - Depth distillation: depth images rendered from per-scene NeRF are used to supervise our model. These per-scene optimized depth images are high-quality, dense, and consistent spatially/temporally
> - Virtual camera distillation: virtual cameras are created as additional targets to artificially increase view overlaps
> - Two-stage Lift-Splat-Shoot strategy: proposed to capture more nuanced depth (Line 126)
> - Features from 2D pre-trained encoder: help the model learn better depth estimations and 2D image features [Line 124]
>
> 2. _Difficulty in processing and coordinating distant/nearby objects in unbounded scene_: Unlike object-centric indoor problems, in the unbounded driving scene, the images usually contain unevenly distributed visual information based on distance: nearby objects occupy significantly more pixels than those far away, even if their physical sizes are identical. This is usually not the case in common NeRF benchmarks, and motivates multiple key designs:
> - Parameterized space: introduced to account for the unbounded scene, unlike SelfOcc/UniPAD with a limited range (~50m) and losing geometry information for far-away scenes
> - Density complement: far-away objects occupy pixels, and thus sampled rays could easily miss them (e.g. distant cars). Thus we propose to query densities from both fine/coarse voxel, and complement density. (Line 175-179 and Fig 3).
> - Light-weight upsampling decoder: applied to rendered feature images, to 1) upsample the final RGB image without additional rendering cost; 2) enable robustness to noises in rendered features
>
> Our paper showed ablation studies on depth distillation, virtual camera distillation, and parameterized space, as in Table 1 and Figure 5. We now additionally provide a qualitative comparison to highlight their differences (see PDF in the joint response). During rebuttal, we also launched ablation studies on all other designs mentioned above, but the training is not finished due to limited time and compute constraints. We will update the results as soon as they are finished, and also in the paper.
>
> > Q2: The differences between the proposed pipeline and other image-based rendering pipelines (GeoNeRF/FeatureNeRF) need clarification.
>
> _Common differences:_
> - GeoNeRF/FeatureNeRF focus on (single) object-level indoor problems and datasets, which are simpler and have less uncertainty since a large number of overlapping views toward the object are available. We target scene-level outdoor reconstruction, a more complex task with sparse non-overlapping views.
> - We introduce distillation from per-scene optimized NeRF to enhance depth/geometry
> - We conduct multi-view fusion to generate a 3D voxel grid
>
> _Specific differences:_
> - GeoNeRF does not leverage the rich information from 2D foundation models, and does not explore downstream tasks
> - FeatureNeRF only considers a single image as inputs
>
> > Q3: advantages of multi-view fusion/voxel grid compared to 3D-to-2D feature projection?
>
> The 3D-to-2D feature projection is exploited in two threads of generalizable approaches: 1) image-based methods, such as GeoNeRF, IBRNet; 2) voxel-based methods, such as UniPAD, NeRF-Det. Due to page limit, we briefly discuss the image-based methods here, but are happy to further elaborate in the discussion stage.
>
> In the image-based generalizable methods, for one query point along the ray from novel views, features are extracted from the feature volume of surrounding source views via projection and interpolation. Our multi-view fusion/voxel grid approach possessed multiple advantages:
> - Explicit Representation: Voxel-based methods provide a direct/explicit 3D scene representation, allowing more straightforward manipulation, analysis, and understanding of spatial relationships in the scene (e.g. removing or replacing object for simulation use)
> - Scalability: As in our case, voxel-based methods can scale to different levels of detail by adjusting the voxel resolution. This scalability allows for efficient representation and rendering of both large scenes and fine details.
>
> > Q4: efficiency aspect.
>
> Please refer to the joint response.
>
> > Q5:  whether the proposed method can be applied to additional common NeRF benchmarks
>
> Our method targets online autonomous driving tasks, with different challenges compared to common NeRF benchmarks as mentioned above. We propose multiple key techniques to address them, and conduct extensive experiments to compare with many related SOTA works in multiple tasks. While we agree that it would be interesting to explore other domains as well, considering the limited time and compute constraints during rebuttal, we were unfortunately not able to include new results in this direction. However, we do believe that our key insights and designs (e.g., distillation from per-scene NeRF and integrating sparse hierarchical voxels with multiple techniques) have a good chance to enhance common NeRF methods and other tasks outside of autonomous driving.

---

> > ### Comment · Reviewer_T1bA · 2024-08-11
> >
> > Thank you to the authors for providing the detailed response. i will keep my rating and listen to our reviewers' opinions.

---

> ### Author Response · Authors · 2024-08-07
> **Settings for the under-training ablation studies.**
>
> | Two-stage LSS | Pretrained encoder | Density complement | Decoder | Metrics |
> |:-------------:|:------------------:|:------------------:|:-------:|:-------:|
> |       ✗       |         ✓          |         ✓          |    ✓    |         |
> |       ✓       |         ✗          |         ✓          |    ✓    |         |
> |       ✓       |         ✓          |         ✗          |    ✓    |         |
> |       ✓       |         ✓          |         ✓          |    ✗    |         |
> |       ✓       |         ✓          |         ✓          |    ✓    |         |

---

> ### Author Response · Authors · 2024-08-07
> **Detailed comparison of our DistillNeRF with GeoNeRF and FeatureNeRF**
>
> |                | Reconstruction  | Offline NeRF Distillation | Multi-View Fusion | Multi-view inputs | Found. Model Lifting | Downstream Task |
> |:--------------:|:---------------:|:-------------------------:|:-----------------:|:------------------:|:--------------------:|:---------------:|
> |    GeoNeRF     |  Object level   |            ✗              |         ✗         |         ✓          |          ✗           |        ✗        |
> |  FeatureNeRF   |  Object level   |            ✗              |         ✗         |         ✗          |          ✓           |        ✓        |
> |     Ours       |  Scene level    |            ✓              |         ✓         |         ✓          |          ✓           |        ✓        |

---

> ### Author Response · Authors · 2024-08-11
> **Update on the requested ablation studies**
>
> Dear reviewer, may we first extend heartfelt gratitude for spending the time reviewing and engaging in the discussion stage, it never goes unappreciated.
>
> As promised, now we are happy to present the ablation studies requested, as in the table below.
>
> | Density complement | Decoder | Pretrained encoder | Two-stage LSS | Depth Distillation | PSNR  | SSIM  |
> |:------------------:|:-------:|:------------------:|:-------------:|:------------------:|:-----:|:-----:|
> | ✗                  | ✓       | ✓                  | ✓             | ✓                  | 22.76 | 0.669 |
> | ✓                  | ✗       | ✓                  | ✓             | ✓                  | 25.34 | 0.839 |
> | ✓                  | ✓       | ✗                  | ✓             | ✓                  | 21.35 | 0.536 |
> | ✓                  | ✓       | ✓                  | ✗             | ✓                  | 27.40 | 0.859 |
> | ✓                  | ✓       | ✓                  | ✓             | ✗                  | 28.01 | 0.872 |
> | ✓                  | ✓       | ✓                  | ✓             | ✓                  | 30.11 | 0.917 |
>
> Specifically, we ablated key components of our sparse voxel representation such as density complement, decoder, pre-trained 2D encoder, and the two-stage LSS. We remove one component each time to ablate its effect. In the last row, we also further add the depth distillation from offline NeRF, which represents the best performance of our full model.
> - No density complement: we observe a significant drop of PSNR from 28.01 to 22.76, demonstrating the importance of better coordination between low-level and high-level sparse voxels.
> - No decoder: we see a decent drop of PSNR from 28.01 to 25.76, showing the effectiveness of using the decoder for robustness to noises in rendered features.
> - No pre-trained 2D encoder: we see a significant drop of PSNR from 28.01 to 21.35, which is expected since using pre-trained 2D encoder has been a commonly acknowledged approach in the field, which SOTA methods such as UniPAD and SelfOcc all adopt.
> - No two-stage LSS: we observe a slight drop of PSNR from 28.01 to 27.40.
> - Add depth distillation from offline NeRF: we observe the jump of PSNR from 28.01 to 30.11.
>
> Note that every ablation above outperforms the SOTA methods UniPAD and SelfOcc, which have PSNR of 19.44 and 20.67 respectively. Among the above techniques, 1) density complement, 2) rendering decoder, 3) two-stage LSS, and 4) distillation from offline NeRF are the novel and unique methods proposed by our paper. With these ablations and analysis for each technique, we believe the key message and contribution of our paper to the community are much clearer.
>
> Finally, the authors want to take a moment to acknowledge your strong expertise and foundations in the field. Your comments are among the deepest and the most constructive, which effectively helps us improve our paper. We acknowledge your precious time spent on the review and discussion.

---

### Author Rebuttal · Authors · 2024-08-07

We sincerely thank all the reviewers for the recognition of our work and the constructive feedback!

The majority of reviewers recommended accepting our work, and evaluated the method to “exhibit good soundness” (R-T1bA), “presents some SotA metrics” (R-WxYD), is “extensively evaluated on NuScene, and achieves very good results for multiple tasks” (R-aj67), and the paper is “interesting to read and simple to follow” (R-aV9Y), “technical details are well elaborated” (R-T1bA), “well-structured and clearly explains the methodology” (R-aV9Y).

The reviewers also raised some questions and suggested improvements. We respond to common points below, and will reply to specific questions in individual replies.

> To respond to questions about reproducibility and implementation details, we would like to share our source code
- Following the rebuttal instructions, no external link is attached and the code is shared with the AC. We’ll also open-source our code along with trained model weights upon the acceptance of the paper.

> We evaluated/analyzed the inference time of our method, along with other comparable methods: (R-T1bA, R-WxYD, R-aj67, R-aV9Y)
* As in the breakdown table below, our model takes 0.4867s for inference, out of which 0.3594s for predicting the voxel from 6 image inputs, and 0.1273s for rendering 6 images (~47 fps for a single camera)

| Component                                | Run Time (s)      |
|------------------------------------------|-------------------|
| Forward inference                        | 0.48672           |
| &emsp;&emsp;Encoder                            | &emsp;&emsp;0.35940     |
| &emsp;&emsp;&emsp;&emsp;Single-view encoding   | &emsp;&emsp;&emsp;&emsp;0.04078   |
| &emsp;&emsp;&emsp;&emsp;Multi-view fusion      | &emsp;&emsp;&emsp;&emsp;0.31862   |
| &emsp;&emsp;&emsp;&emsp;&emsp;&emsp;Voxel convolution  | &emsp;&emsp;&emsp;&emsp;&emsp;&emsp;0.2494   |
| &emsp;&emsp;Renderer               | &emsp;&emsp;0.12730   |
| &emsp;&emsp;&emsp;&emsp;Projection + Ray march | &emsp;&emsp;&emsp;&emsp;0.12646   |
| &emsp;&emsp;&emsp;&emsp;Decoder                            | &emsp;&emsp;&emsp;&emsp;0.00086     |


* In our paper, we demonstrated that our method significantly outperforms other SOTA generalizable methods SelfOcc and Unipad (e.g. reconstruction PSNR of 30.11, 20.67, 19.44, respectively). In terms of inference speed, SelfOcc and UniPad have total inference times of 0.1771s and 0.6514s, respectively, compared to our 0.4867s.
     - SelfOcc is expected to be fast since it adopts an implicit representation where deformable cross-attention is used to aggregate information from the image features to generate a 3D SDF field. In comparison, our explicit voxel representation takes decent time for the 3D convolution operations, but offers additional benefits such as more straightforward manipulation (e.g. removing or replacing object for simulation use), and flexibility to scale to different levels of detail by adjusting the voxel resolution.

     - UniPAD adopts a voxel-based representation that is similar and more comparable to our method, while our method shows faster inference speed, presumably due to key designs such as voxel sparsification, and the lightweight decoder that enables efficient rendering.

- The evaluation is conducted on the same desktop-grade machine (13th Gen Intel(R) Core(TM) i7-13700KF, NVIDIA GeForce RTX 4090) and rendering the same image resolution (228*128)

> We conducted more ablation studies to better understand each component in our model, and clarified details in existing ablation studies (R-T1bA, R-aj67)
- In response to R-T1bA: we highlighted existing ablation studies (see the attached PDF), and conducted more comprehensive ablation studies of key components of our model, so that the key messages and effects of each component are clearer and easier to refer to for people in the community. While the training of these ablation studies is not finished yet due to limited time and compute constraints, we will update the results as soon as the training is finished (e.g. during the discussion stage), and also integrate them into the final paper. We refer to the response to R-T1bA for more detailed descriptions of these ablations.
- In response to R-aj67, we clarified the notations and improved the formats of existing ablation studies for clarity
- In response to R-aj67, we created another baseline for generating foundation model feature images, and demonstrated that our approach possesses significantly faster inference speed (x3.3 and x1.89 times faster).

---

### Decision · Program_Chairs · 2024-09-25

**Decision:**

Accept (poster)

**Comment:**

The paper received four reviews from experts in the field. After the rebuttal and discussion phases, the final scores remain mixed. The rebuttal cleared up most open questions, added timings, additional ablations, and further clarifications.
Reviewer WxYD main remaining concern is that the promised changes will be included in the final version. The AC has read the paper, the reviews, the rebuttal, and the discussion and finds that the explanations provided in the rebuttal are sufficient evidence for inclusion in the paper. The author-reviewer discussion period is the exact mechanism to enable reviewers and authors to discuss issues of clarity and missing details. This involves a certain level of trust but is further encouraged by the fact that reviews, meta-reviews, and any discussion with the authors will be made public for accepted papers.

The AC thus recommends acceptance of the paper with the urge to include all necessary details to replicate the method in the text and not only in the source code.